# Genomic insight into domestication of rubber tree

Jinquan Chao[1,2,3,15], Shaohua Wu[1,2,3,15], Minjing Shi[1,2,3,15], Xia Xu[4,15], Qiang Gao[5,6,15], Huilong Du[4,7,8,15], Bin Gao[9], Dong Guo[10], Shuguang Yang[1,2,3], Shixin Zhang[1,2,3], Yan Li[1,2,3], Xiuli Fan[4,8], Chunyan Hai[4,8], Liquan Kou[4], Jiao Zhang[11], Zhiwei Wang[11], Yan Li[11], Wenbo Xue[11], Jiang Xu[12], Xiaomin Deng[1,2,3], Xiao Huang[2,3], Xinsheng Gao[1,2,3], Xiaofei Zhang[1,2,3], Yanshi Hu[1,2,3], Xia Zeng[1,2,3], Weiguo Li[1,2,3], Liangsheng Zhang[5], Shiqing Peng[10], Jilin Wu[2,3], Bingzhong Hao[2,3], Xuchu Wang[13], Hong Yu[4], Jiayang Li[4,8,14 ✉], Chengzhi Liang[4,8 ✉] & Wei-Min Tian[1,2,3 ✉]

Understanding the genetic basis of rubber tree (*Hevea brasiliensis*) domestication is crucial for further improving natural rubber production to meet its increasing demand worldwide. Here we provide a high-quality *H. brasiliensis* genome assembly (1.58 Gb, contig N50 of 11.21 megabases), present a map of genome variations by resequencing 335 accessions and reveal domestication-related molecular signals and a major domestication trait, the higher number of laticifer rings. We further show that *HbPSK5*, encoding the small-peptide hormone phytosulfokine (PSK), is a key domestication gene and closely correlated with the major domestication trait. The transcriptional activation of *HbPSK5* by myelocytomatosis (MYC) members links PSK signaling to jasmonates in regulating the laticifer differentiation in rubber tree. Heterologous overexpression of *HbPSK5* in Russian dandelion (*Taraxacum kok-saghyz*) can increase rubber content by promoting laticifer formation. Our results provide an insight into target genes for improving rubber tree and accelerating the domestication of other rubber-producing plants.

Plant domestication is a genetically dynamic and ongoing process of a wild species to adapt to human use by selecting specific traits[1,2]. Knowledge about the genetic basis of plant domestication mainly comes from annual crops[3,4]. It appears that the domestication process of long-lived perennials differs dramatically from that of annuals in that perennials tend to be domesticated at a slower rate, are typically out-crossing, have extended juvenile periods, are propagated clonally, and exhibit significantly fewer domestication syndrome traits[1,2]. The dynamics of domestication in perennials differs markedly from that of annuals, and the genetic basis of domestication in perennials has been studied in less depth[2].

*Hevea brasiliensis* (rubber tree) is a perennial crop of Euphorbiaceae and indigenous to the Amazon Basin in South America[5]. The domestication of rubber tree has contributed greatly to the human lifestyles in the past 150-year[6,7]. The rubber tree provides more than 98% of the total natural rubber (NR) production worldwide[8] although there are ~2000 rubber-producing plant species including *Parthenium argentatum*[9], *Taraxacum kok-saghyz*[10], and *Eucommia ulmoides*[11]. The NR is an essential material, especially for plane and truck tyres and medical gloves[9]. The rubber tree was primarily wild-harvested in South America before Henry Wickham collected rubber tree seeds in Boim of Brazil and introduced them into Kew Botanic Garden of the United Kingdom, starting the domestication process of rubber tree in 1876 (refs. 6, 12). So far most of the *Hevea* cultivars around the world are domesticated from Wickham collections. To widen the genetic basis for generating new clones with regarding to productivity, disease

---

resistance, and tolerance to many environmental conditions, the International Rubber Research and Development Board (IRRDB) organized experts to collect wild Amazonian rubber tree germplasms from several districts of the states of Acre, Rondonia, and Mato Grosso of Brazil in 1981, termed as 1981' IRRDB wild germplasms[5].

The domestication process of rubber tree generally fits in with the dynamics of domestication in perennial crops[2], but differs from apple tree that interspecific hybridization plays a major role during domestication[13]. The domestication of rubber tree starts directly from Wickham collections without interspecific hybridization. The difficulty to dissect the component traits of yield as well as the long asexual period, however, make the traditional breeding process of rubber tree time-consuming (>20–25 years)[14], resulting in less than four times of manual guidance sexual recombinations in *Hevea* cultivars within 150 years to date. Nevertheless, the individual with favorite traits can easily be retained and propagated by bud grafting. In this way, the domestication has led to a multifold increase in NR yield from 650 kg rubber ha$^{-1}$ of the haphazardly selected germplasms during the 1920s to 2500 kg ha$^{-1}$ of the elite cultivars by the 1990s[7]. The obviously improved productivity of rubber tree cultivars suggests the likely presence of major component loci closely related to NR yield because the few occurrences of sexual recombinations were unlikely to be able to pyramid numerous minor loci for increasing yield.

The latex is essentially the cytoplasm of laticifer cells that contains 30–40% NR (*cis*-1,4-polyisoprene) and is exploited by tapping, a process of cutting laticifer rings in a non-destructive manner to facilitate continual latex production for many years[15]. The tapping-caused latex exploitation enhances, in turn, latex regeneration[16] and laticifer formation[17]. The articulated laticifer in rubber tree is composed of laticifer cells that are specific for NR biosynthesis and storage. Their synchronous differentiation from the fusiform initials of vascular cambia results in a circular arrangement of laticifer cells[18]. Each circle of laticifer cells is considered a laticifer ring. The number of laticifer rings (NLR) is a major component trait of rubber yield, given that NLR is positively correlated with rubber yield[19].

In this work, we assemble a high-quality genome of the rubber tree clone CATAS8-79, re-sequence 335 accessions and perform a population genetic analysis and genome wide association study (GWAS) based on the map of genome variations and a major domesticated trait NLR. The revealed domestication-related molecular signals and one identified key domestication-related gene will be valuable for improving rubber tree and accelerating the domestication of other rubber-producing plants and possibly, perennial crops.

## Results

### Genome assembly, annotation, and comparative genomic analyses

The elite rubber tree cultivar CATAS8-79 (a rubber tree clone selected by Chinese Academy of Tropical Agricultural Sciences) was selected for genome sequencing and assembling. The genome size was estimated to be 1.50–1.62 Gb based on the 17-mer to 27-mer distribution analysis (Supplementary Table 1). By integration of 192 Gb of PacBio CLR (continuous long-read) sequences, 110 Gb high-throughput chromatin conformation capture (Hi-C) data, and 206.85 Gb of BioNano data (Supplementary Table 2), we finally achieved a 1.58 Gb assembly (1.55 Gb anchored on 18 pseudochromosomes) with a contig N50 of 11.21 Mb and largely reduced fragments (Table 1, Fig. 1, Supplementary Table 3 and Supplementary Data 1) in comparison to the previous versions[20–24]. The assembly quality and completeness were validated by four approaches. Firstly, 85.58 Gb next-generation sequencing (NGS) clean reads were mapped to the assembly, yielding 97.74% mapping ratio. Secondly, we identified 1546 (95.8%) of the 1614 conserved embryophyta proteins collected from the Benchmarking Universal Single-Copy Orthologs (BUSCO) program, which is higher than 1509 (93.50%) of rubber tree clone GT1 assembly (Table 1).

**Table 1 | Statistics of rubber tree clone CATAS8-79 and GT1 genome assemblies**

| | CATAS8-79 | GT1 (Liu et al.[20]) |
|---|---|---|
| **Sequence statistics** | | |
| Next-generation sequencing (Gb) | 85.58 | 348.14 |
| PacBio (Gb) | 192.00 | 161.86 |
| Hi-C (Gb) | 110.00 | 119.63 |
| BioNano (Gb) | 206.85 | n.a. |
| **Feature of genome assembly** | | |
| Total length of assembly (Gb) | 1.58 | 1.56 |
| Total no. of contigs | 1058 | 16,023 |
| Contig N50 (Mb) | 11.21 | 0.15 |
| Longest contig (Mb) | 56.83 | 1.81 |
| Total no. of scaffold | 776 | 600 |
| Scaffold N50 (Mb) | 94.82 | 84.02 |
| Longest scaffold (Mb) | 112.33 | 103.98 |
| Total no. of anchored contigs | 300 | 15,324 |
| Gap number | 282 | 15,306 |
| Total length of chromosome-level assembly (Gb) | 1.55 | 1.44 |
| Anchor rate | 98.17% | 92.40% |
| Mercury completeness | 82.11% | 82.37% |
| Mercury quality value | 28.92 | 24.62 |
| Complete BUSCOs | 1546 (95.80%) | 1509 (93.50%) |
| Complete and single-copy BUSCOs | 1377 (85.30%) | 1349 (83.60%) |
| Complete and duplicated BUSCOs | 169 (10.50%) | 160 (9.90%) |
| Fragmented BUSCOs | 34 (2.10%) | 44 (2.70%) |
| Missing BUSCOs | 34 (2.10%) | 61 (3.80%) |
| Total BUSCO groups searched | 1614 (100.00%) | 1614 (100.00%) |
| **Gene annotation** | | |
| No. of gene models | 38,595 | 44,187 |
| Percentage of gene length in genome | 13.15% | 11.10% |
| Max gene length (bp) | 75,969.00 | 72,122.00 |
| Average gene length (bp) | 5282.54 | 3918.40 |
| No. of transcripts | 82,641 | 102,235 |
| Max transcript length (bp) | 21,964.00 | 10,920.00 |
| Average transcript length (bp) | 1680.61 | 1065.00 |
| Average CDS length (bp) | 1275.52 | 1140.10 |
| Average exon length (bp) | 217.09 | 222.08 |
| Average intron length (bp) | 552.46 | 672.1 |
| Average exon number per gene | 7.74 | 5.13 |
| Repeat sequence length (Mb) | 1180.12 | 1042.40 |
| Repeats percentage of assembly | 75.92% | 70.81% |

The slightly higher ratio of duplicated genes in rubber tree might be due to the WGD events[20,21]. Thirdly, the long terminal repeat (LTR) assembly index (LAI) score of the present assembly reached to 13.67, which was significantly higher than the 8.91 of rubber tree clone GT1 assembly (Table 1). Lastly, the mercury completeness and consensus quality value (QV) were analyzed with much higher value of QV in CATAS8-79 assembly than that in GT1 assembly (Table 1).

A total of 1.18 Gb was identified as repeats including 929.11 Mb LTR-retrotransposons, which was the highest in comparison to the published rubber tree genomes (Table 1, Supplementary Table 4 and Supplementary Data 1). The gene annotation was performed by combining de novo gene prediction, homology searches, Iso-Seq and RNA-Seq of seven tissues (Supplementary Data 2). A total of 38,595 high-confidence gene models were predicted and functionally annotated in public databases (Table 1 and Supplementary Table 5). The intra-

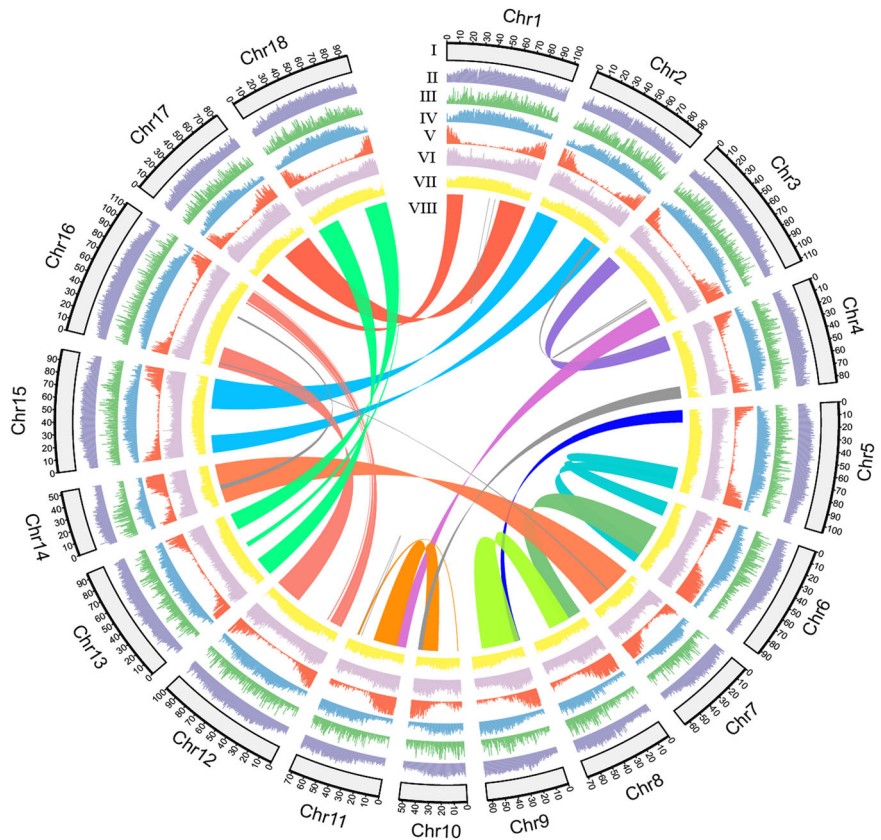

**Fig. 1 | Features of rubber tree clone CATAS8-79 genome with an assembly size of 1.58 Gb and a contig N50 of 11.21 Mb. I** 18 chromosomes; **II** density transposon elements in a 100-kb sliding window; **III** density of *copia* retrotransposons in a 500-kb sliding window; **IV** density of *gypsy* retrotransposons in a 500-kb sliding window; **V** gene density in a 500-kb sliding window; **VI** gene expression values represented by the highest expression level in any of seven tissues based on log2(FPKM+1) of RNA-seq data; **VII** distribution of guanine-cytosine content in a 500-kb sliding window, average GC content reached to 34.62%; **VIII** Intragenomic syntenic regions covering at least 20 paralogues. Source data are provided as a Source data file.

genome syntenic analysis identified 641 syntenic blocks in CATAS8-79 (Supplementary Fig. 1a) while 1901 in GT1 (Supplementary Fig. 1b). The inter-genome syntenic analysis showed that the number of syntenic blocks (1235) between CATAS8-79 and GT1 genotypes was much more than that (570) between CATAS8-79 and cassava (Supplementary Fig. 1c).

Gene gain-loss analysis of rubber tree and 11 other species showed that 401 families exhibited expansion while 1554 families appeared contraction in rubber tree (Supplementary Fig. 2, Supplementary Table 6 and Supplementary Data 3). In addition to the expansion of small rubber particle protein/rubber elongation factor (SRPP/REF) family (Supplementary Data 4) as reported previously[20,21], we also detected a remarkable expansion of the myelocytomatosis (MYC) transcription factor family in rubber tree. A total of 26 *HbMYCs* were identified from *Hevea brasiliensis* (Supplementary Fig. 3a, Supplementary Data 4 and 5). Of which, 16 members were located on three chromosomes, Chr5, Chr6, and Chr8 (Supplementary Fig. 3b). The clustered arrangement of *HbMYCs* was like that of the rubber biosynthesis-related *SRPP/REF* genes in rubber tree[20,21] and the morphine biosynthesis-related genes in opium poppy[25]. Phylogenetic analysis showed that the MYCs from different plant species could be divided into seven groups (Supplementary Fig. 3c and Supplementary Data 5). An obvious expansion occurred in group IV, which was mainly composed of members from cassava and rubber tree (Supplementary Fig. 3c) and driven by whole genome duplication events (Supplementary Fig. 3d). Notably, five HbMYCs in group VII were clustered together with the MYC members from *Taraxacum kok-saghyz* (TKS) (Supplementary Fig. 3c), a rubber-producing plant belonging to Asteraceae[10]. The MYC members in group VII are likely associated with

the regulation of laticifer formation and rubber biosynthesis, considering the similarity in laticifer structure and function between the two species that were located far away taxonomically[10]. Most of *HbMYCs* were notably up-regulated in laticifer cells by tapping and in the cambial region upon treatment with coronatine (COR), a jasmonic acid mimic (Supplementary Fig. 3e).

## Population genomic analyses revealed domestication clues to yield selection

A total of 335 accessions including 127 cultivars and 208 wild germplasms of *H. brasiliensis* together with one germplasm of *H. spruceana* were re-sequenced and used for population genomic analysis (Fig. 2a and Supplementary Data 6). After aligning data against the present assembly, 5,323,701 SNPs and 259,445 InDels were generated (Supplementary Table 7).

A total of 336,200 SNPs located in coding sequence regions were used to construct a phylogenetic tree of 335 accessions of *H. brasiliensis* with *H. spruceana* as the outgroup. Accessions were obviously grouped into four major clades (Fig. 2b), including cultivars (Cul) and wild accessions from Acre (WAC), from Rondonia (WRO), and from Mato Grosso (WMG) (Supplementary Data 6). Population structure analysis confirmed that K= 4 represented the best model for diverging the four populations (Fig. 2c and Supplementary Fig. 4). It was noted that there were two subclusters in clade WRO at K = 10 (Fig. 2c). The two subclusters, WRO1 and WRO2, were caused by the difference in geographic location with *Fst* value of 0.089 (Fig. 2c and Supplementary Table 8). The population structure was further supported by principal component analysis (PCA, Fig. 2d). We noticed that the fixation index (*Fst*) value was only 0.07 between clades WMG and Cul, which was

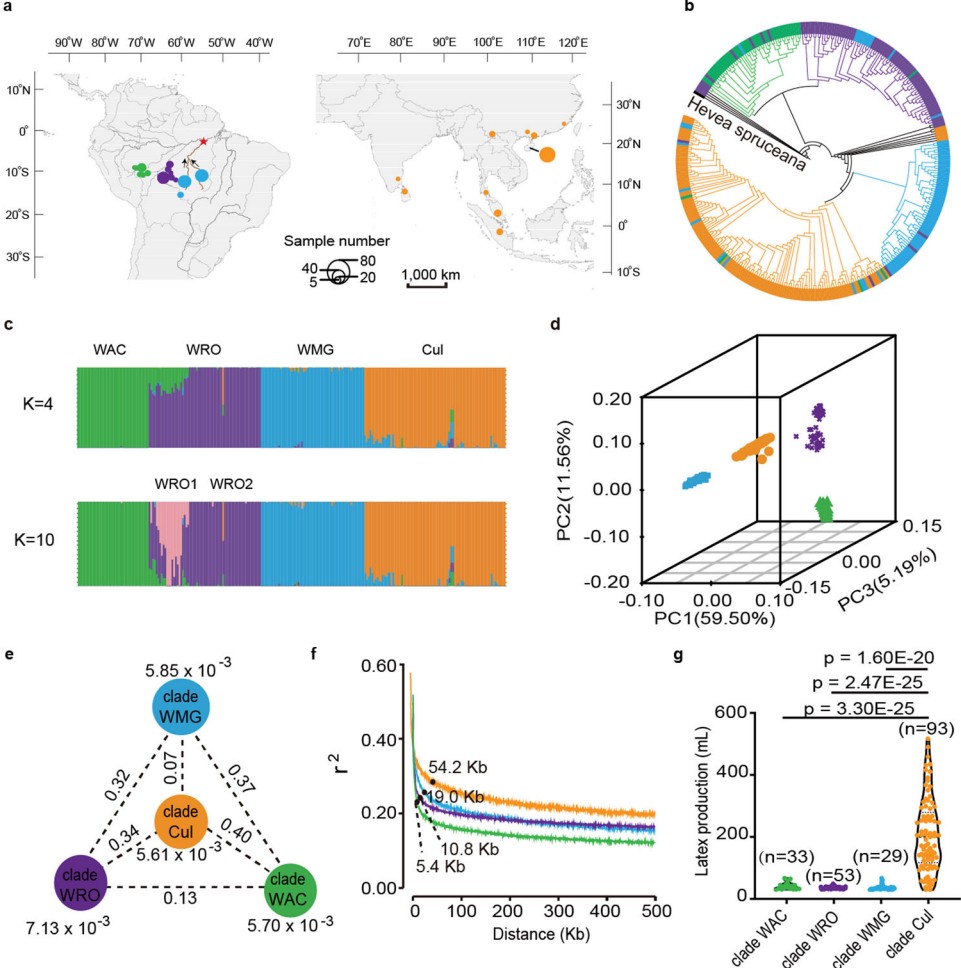

**Fig. 2 | Geographic distribution and population analysis of 336 *Hevea* accessions. a** Collection places of wild and cultivated *Hevea* accessions. The circles in different diameter represented the different number of collected accessions. The star indicated the location from where the Wickham germplasms were collected. Green, purple, and blue cycles represented the wild germplasms from Acre, Rondonia, and Mato Grosso, respectively. The yellow cycles represented the cultivars. **b** Phylogenetic tree of *Hevea* accessions with *H. spruceana* as an outgroup. The different colors of the circular perimeter indicated the group name of each clade. **c** Structure analysis with K = 4 and 10 with the *Fst* values between groups. **d** PCA plot of the first three components (PC1, PC2, and PC3). **e** Population divergence and genetic diversity across the four clades. The values in the nodes represented the genetic diversity (π) of the clades (yellow, blue, green, and purple nodes represented clade Cul, clade WMG, clade WAC, and clade WRO clades, respectively), and the values between the nodes indicated population divergence (*Fst*). **f** Estimation of LD decay-distance of the four clades. **g** Latex production of the four clades. The results are expressed as means ± s.d. in three biological replicates. The number of accessions for each clade is shown in figure. *P* values are calculated with two-sided Student's *t*-test. Source data are provided as a Source data file.

significantly lower than the *Fst* values between clades WMG and WAC (0.37) and between clades WMG and WRO (0.32) (Fig. 2e). These findings indicate that the three wild populations had diverged to some extent, and the cultivars were derived from clade WMG, which strengthens the result of the previous report[26].

The nucleotide diversity (π) of rubber tree cultivars ($5.61 \times 10^{-3}$) was not significantly lower in comparison to that of the wild germplasms ($5.70 \times 10^{-3}$–$7.13 \times 10^{-3}$, Fig. 2e), which was different from the remarkable decrease in genetic diversity in seed-propagated annual crops[27]. The little reduction in genetic diversity of rubber tree cultivars should be mainly ascribed to the bud grafting propagation, which could contribute to retain the genetic diversity in many perennial crops[2]. In addition, the fewer occurrences of sexual recombinations and the large size and high heterozygosity of the rubber tree genome may also help to maintain genetic diversity. Nevertheless, the greater linkage disequilibrium (LD) distance in clade Cul (54.2 kb) than in clade WAC, WRO, and WMG (5.4–19.0 kb, Fig. 2f) suggests that there was artificial selection during domestication of rubber tree within 150 years. The latex production of cultivars was significantly higher than that of wild relatives (Fig. 2g and Supplementary Data 7).

To identify domestication-related molecular signals, selective sweep analysis was performed between clades Cul and WMG (Fig. 3a). A total of 21.58, 22.72, and 22.56 Mb of genomic sequences covering 1688, 1661, and 1804 genes were detected using the *Fst* (Supplementary Data 8), the π (Supplementary Data 9), and the XP-CLR (Supplementary Data 10) methods, respectively. Of the genes identified by the three methods, we focused on those related to jasmonate (JA) and ethylene signaling, rubber biosynthesis and reactive oxygen species (ROS) metabolism in consideration of the role of JA signaling in regulating laticifer differentiation[16,18,19] and rubber biosynthesis[17] and of ethylene signaling and ROS metabolism in regulating the duration of latex flow[28]. Selective sweeps were observed in 11 JA signaling-related genes (*HbLOX17*, *HbLOX18*, *HbAOC2*, *HbOPR2*, *HbOPR8* and *HbOPR9*, *HbOPR10*, *HbMYC24*, *HbHAT3*, *HbMAPKKK35*, and *HbMAPKK2*), nine rubber biosynthesis-related genes (*HbNIN3*, *HbSUT5*, *HbACAT4*, *HbMVD1*, *HbFPS1*, *HbGPS1*, *HbGPS4*, *HbSRPP4*, and *HbREF1*), thirteen ethylene signaling-related genes (*HbACS4*, *HbACO25*, *HbEIN4*, *HbERF2*, *HbERF19*, *HbERF20*, *HbERF25*, *HbERF26*, *HbERF48*, *HbERF61*, *HbERF100*, *HbERF142*, and *HbERF145*) and nine ROS scavenging-related genes (*HbSOD4*,

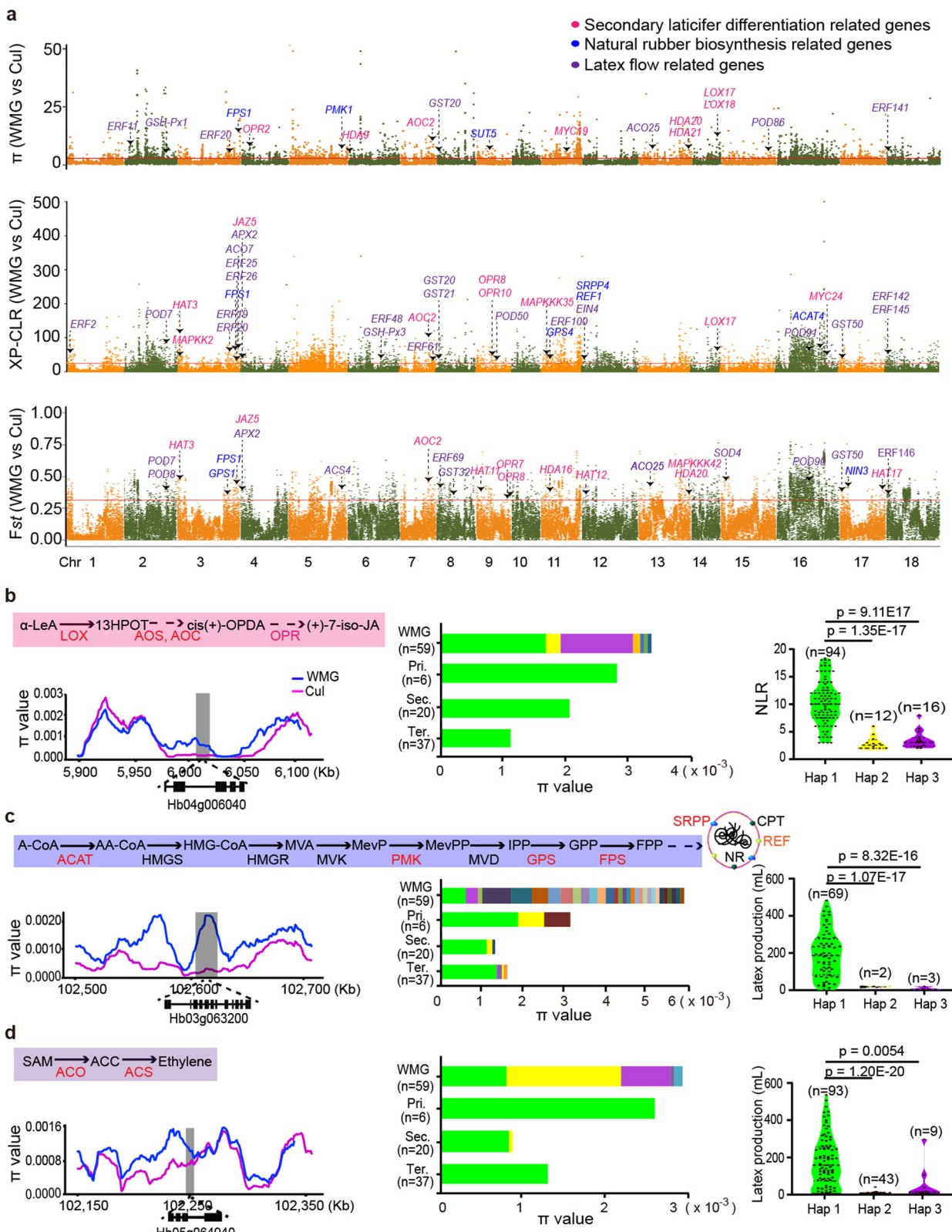

**Fig. 3 | Genome-wide distribution of selective sweeps on the CATAS8-79 genome and selective features. a** Selective sweeps detection by *Fst* and π methods. **b–d** Changes in nucleotide diversity (π) and enrichment of haplotypes of *HbOPR2* (**b**), *HbFPS1* (**c**), and *HbACS4* (**d**) across three generations and the correlations of haplotypes with corresponding phenotypes (NLR and latex production). Pri,

primary clones; Sec, secondary clones; Ter, ternary clones; Hap, haplotype. n, the number of accessions. The results are expressed as means ± s.d. in three biological replicates. The number of accessions for each haplotype is shown in figure. *P* values are calculated with two-sided Student's *t*-test. Source data are provided as a Source data file.

*HbPOD7, HbPOD50, HbPOD91, HbGST20, HbGST21, HbGST50, HbGSH-Px3,* and *HbAPX2*) (Fig. 3a).

These selected genes might confer the higher NLR and rubber biosynthesis efficiency and longer duration of latex flow in cultivars in comparison with the wild relatives (Supplementary Fig. 5). The expression of *HbOPR2*, encoding a homolog of AtOPR3 that is a key enzyme for endogenous JA biosynthesis[29], was rapidly up-regulated in the cambia region in response to exogenous coronatine, which may be associated with the biosynthesis of endogenous JA and the secondary laticifer differentiation[30] (Supplementary Fig. 5a, b). The expression levels of *HbNIN3* and *HbFPS1* were remarkably up-regulated (Supplementary Fig. 5c), which may be associated with the enhanced latex regeneration in regularly tapped tree[17]. The *HbACS4* was mainly up-regulated in the inner bark of tapped trees (Supplementary Fig. 5d), which appears to confer the endogenous ethylene production upon tapping, resulting in a prolongation of duration of latex flow. A higher transcriptional level of *HbSOD4* in the rubber tree clone CATAS8-79 (Supplementary Fig. 5e) may be associated with the increased duration of latex flow in comparison with rubber tree clone PR107 (ref. 31).

Analysis of nucleotide diversity and haplotype of the above-mentioned genes was performed, and representatives were presented (Fig. 3b–d). The π values of *HbOPR2, HbFPS1,* and *HbACS4* were notably lower in cultivars in comparison to that in the wild germplasms from Mato Grosso, and the π values further declined with subsequent generation of cultivars (Fig. 3b–d). We also noted that a preferential haplotype in *HbOPR2, HbFPS1,* and *HbACS4* was enriched in the primary clones and retained in the next two generations, positively correlating with NLR and latex production, respectively (Fig. 3b–d and Supplementary Data 11).

## Identification and functional verification of a key domestication gene

We evaluated the NLR of 93 cultivated and 115 wild germplasms that had been tapped for three years (Supplementary Data 7). The NLR was notably diverse among the germplasms (Supplementary Fig. 6a). A strong association signal was detected in Chr15 by GWAS (Fig. 4a), including a 20-kb overlapping region with a selective sweep region obtained by the *Fst* method (Fig. 4b). We examined the tissue-specific expression of 32 genes within a 500-kb region around the GWAS signal peak (Fig. 4c and Supplementary Data 12) and found that only *Hb15g001250* was specifically expressed in the inner bark, particularly in the cambial region. Interestingly, *Hb15g001250*, one of the two genes within the 20-kb overlapping region (Supplementary Fig. 6b), encodes the phytosulfokine (PSK), a small-peptide hormone[32]. By scanning the CATAS8-79 genome, six genes potentially encoding for PSKs were identified (Supplementary Table 9). The *Hb15g001250* gene was named as *HbPSK5* based on the order of the gene accession number. Of the six members, *HbPSK5* was also the only one specifically expressed in the inner bark tissues and cambial region (Supplementary Table 9). We found one significantly associated SNP (Chr15:1200714) in the *HbPSK5* promoter region 760 bp upstream of the transcription start site (Fig. 4d). This site had three genotypes: GG (same as the reference genome), AA, and GA (heterozygote). The distribution patterns of the three genotypes were retrieved by resequencing and verified by Sanger sequencing (Supplementary Data 13). The genotype GG was associated with higher NLR while genotype AA the fewer NLR among the evaluated 93 cultivated and 115 wild germplasms (Fig. 4e, Supplementary Data 7 and Supplementary Data 14). This relationship was verified by high percentage of genotype AA (93.27%) and low percentage of genotype GG (0.96%) in the 208 wild germplasms that generally possessed fewer NLR (Fig. 4f and Supplementary Data 15). It was also verified by high percentage of genotype GG (44%) in the germplasms with more laticifer rings (ML) and low percentage of genotype GG (8.33%) in the germplasms with fewer laticifer rings (FL) from hybrid segregation populations (Fig. 4g and Supplementary

Data 15). The genotype GG was selected during domestication. The percentage of genotype GG increased from 9.09% in the primary clones to 20% in the secondary clones and to 37.84% in the ternary clones (Fig. 4h and Supplementary Data 14). Such evidence strongly supports that *HbPSK5* was a key domestication gene and is crucial to the process of laticifer differentiation. The PSK gene family seems to be absent from bryophyte and pteridophyte while present in spermophyte (Supplementary Data 16). The PSK members are more conservative in gymnosperm than in angiosperm while diverge more seriously in monocotyledon than in dicotyledon (Supplementary Fig. 7 and Supplementary Data 17).

The role of PSK in regulating laticifer differentiation was testified by application of synthetic PSK to the epicormic shoots of rubber tree clone CATAS7-33-97. As positive controls, mechanical wounding or 20 μM COR could effectively induce the differentiation of the secondary laticifer cells in the secondary phloem close to the cambia (Fig. 4i). As a negative control, H₂O had no effect on inducing laticifer differentiation (Fig. 4i). Different concentrations of synthetic PSK were respectively applied to the epicormic shoots. The application of 100 μM synthetic PSK was more effective in inducing laticifer differentiation (Fig. 4i). The effect of synthetic PSK was weaker than that of mechanical wounding or COR (Fig. 4i), suggesting a low translation efficiency of exogenous synthetic PSK or the likely presence of other cues in regulating laticifer differentiation. The similar effect of synthetic PSK and COR in inducing laticifer differentiation suggests a close association between PSK signaling and JA signaling. *HbPSK5* had four G-box motifs in its promoter region (Fig. 4d and Supplementary Fig. 6c) and was rapidly up-regulated in the cambial region by the application of COR (Supplementary Fig. 8a). It may be the target gene of MYC transcription factors that act as a core component of JA signaling by binding to the G-box motif of JA-responsive genes[33]. A yeast one-hybrid assay showed that HbMYC10, HbMYC15, HbMYC18, HbMYC25, and HbMYC26 of the 26 HbMYCs interacted with the promoter region of *HbPSK5* (Supplementary Fig. 8b, c). The physical binding of HbMYC18 to the promoter of *HbPSK5* was further verified by an electrophoretic mobility shift assay (EMSA, Supplementary Fig. 8d). Transient dual-luciferase assays in tobacco revealed that each of these five MYC members, especially HbMYC26, could activate the *HbPSK5* promoter (Fig. 4j). The role of *HbPSK5* and *HbMYC26* in laticifer formation was further identified by their heterologously overexpressed in TKS, respectively. The expression level of *HbMYC26* and *HbPSK5* was positively correlated with the laticifer cell index (Supplementary Figs. 9 and 10). Overexpression of *HbMYC26* and *HbPSK5* in TKS promoted laticifer formation, causing 14.29–49.38% and 15.48–76.77% increases in rubber contents, respectively (Fig. 4k–n, Supplementary Figs. 9 and 10).

## Discussion

Domestication of rubber tree has greatly changed modern human civilization and lifestyle. The 150-year domestication provides an opportunity to explore the genetic basis of recent domestication, in contrast to the domestication of typical food crops which undergoes several thousand years. As an industrial crop with little edible value, rubber tree had been primarily wild-harvested in South America before 1876 when domestication began outside of its indigenous regions[6,12]. The population genetic analysis showed that Wickham collections-originated rubber tree cultivars were derived from the Mato Grosso wild germplasms, strengthened by the shared hydrographical network by the places where original Wickham collections and Mato Grosso wild germplasms were collected[26]. Although 150-year domestication did not significantly reduce the nucleotide diversity genomewide, there was artificial selection based on the obviously increased LD distance of cultivars, which might be partially caused by the reduced π value and enrichment of preferential haplotype in yield productivity-related genes by generations.

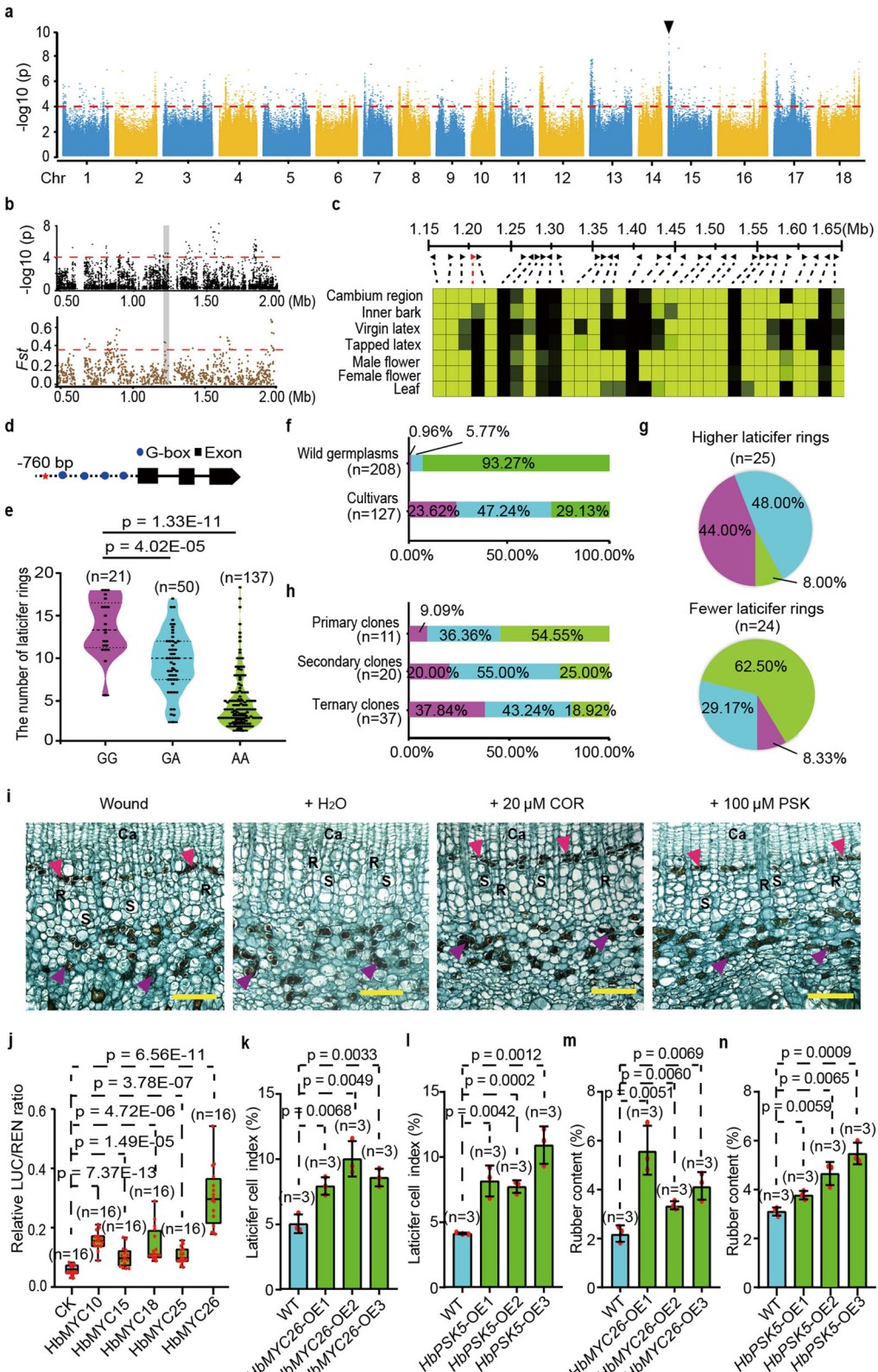

The cultivars generally possessed higher NLR than their wild relatives although artificial selection was conducted by the amount of exploited latex. The inadvertent selection of the higher NLR, in turn, verified a closely association of NLR with rubber yield. Given that laticifer cells are the sites of NR biosynthesis and storage, the higher number of laticifer cells should be a typical domestication syndrome trait of rubber-producing plant species, including the largely

undomesticated TKS[34]. We identified that a phytosulfokine gene, *HbPSK5*, was a domesticated gene. It was positively associated with NLR by promoting the differentiation of vascular cambia into laticifer cells downstream of JA signaling. Overexpression of *HbPSK5* and its transcriptional activator *HbMYC26* respectively caused a significant increase in rubber content by promoting laticifer differentiation in TKS. The *HbPSK5* is so far an identified gene that has a role in

**Fig. 4 | GWAS for NLR and functional analysis of *HbPSK5*. a** Manhattan plot of GWAS for NLR. The black arrowhead indicated a significant signal. **b** Fine mapping of the region related to NLR. The gray column represented an overlapping region of 20 kb. **c** Tissue-specific expression of putative genes around the GWAS signal peak. The bar represented FPKM values. **d** Gene structure of *HbPSK5*. Red star indicated a significant SNP site. **e** Relationships of NLR with genotypes GG, GA, and AA among 93 cultivated and 115 wild germplasms. The results are expressed as means ± s.d. in three biological replicates. The number of accessions for each category is shown in figure. *P* values are calculated with two-sided Student's *t*-test. **f** Distributions of the three genotypes in 127 cultivars and 208 wild germplasms. **g** Distributions of the three genotypes in offspring with more laticifer rings (ML) and fewer laticifer rings (FL) identified from hybrid separation populations. **h** The proportions of the three genotypes differed by generation. **i** Representative light micrographs of bark cross-sections in ten biological replicates, showing the effects of mechanical wounding,

H₂O, COR, and PSK on the differentiation of secondary laticifer cells. Red arrowhead, secondary laticifer cells; pink arrowhead, primary laticifer cells; Ca, cambia; R, ray cell; S, sieve tube. Scale bars = 100 μm. **j** Assays of the transient transcriptional activities of HbMYCs on *HbPSK5*. The results are expressed as means ± s.d. in sixteen biological replicates. In each box plot, the center line indicates the median, the edges of the box represent the first and third quartiles, and the whiskers extend to span a 1.5 interquartile range from the edges. *P* values are calculated with two-sided Student's *t*-test. **k**, **l** Effects of overexpressed *HbMYC26* (**k**) or *HbPSK5* (**l**) on the differentiation of laticifer cells in TKS. The results are expressed as means ± s.d. in three biological replicates. *P* values are calculated with two-sided Student's *t*-test. **m**, **n** Effects of overexpressed *HbMYC26* (**m**) or *HbPSK5* (**n**) on the rubber content in TKS. The results are expressed as means ± s.d. in three biological replicates. *P* values are calculated with two-sided Student's *t*-test. Source data are provided as a Source data file.

promoting the differentiation of vascular cambia into a specific parenchyma tissue in the secondary phloem[35], which deepens understanding of the role of PSK in plant physiological processes[36–39]. In addition, little is known about how JA signaling regulates mitogen-activated protein kinase (MAPK) cascades although the MAPKs are clearly involved in JA signaling pathway[40]. Our results suggest that JA signaling may regulate MAPK cascades by activating PSK signaling. These insights will accelerate the improvement of rubber-producing plants and provide a perspective for small-peptide hormones in regulating vascular cambial differentiation.

## Methods
### Plant materials and treatment
The elite rubber tree cultivar CATAS8-79, bred by the Rubber Research Institute of Chinese Academy of Tropical Agricultural Sciences (RRI-CATAS), was used for de novo genome assembly. Fresh young leaves were harvested and stored under low temperature (4 °C) for genome sequencing, BioNano and Hi-C analyses. Seven types of tissues (cambial region, inner bark, virgin tree latex, tapped tree latex, female flower, male flower, and leaves) were harvested and frozen in liquid nitrogen for RNA extraction and transcriptome sequencing.

A total of 335 *H. brasiliensis* accessions together with one *H. spruceana* accession were preserved in the field of the National Tropical Plants Germplasm Resource Center-Rubber Tree (Danzhou, Hainan Province) and used for whole genome resequencing. With the aid of cultivar pedigrees, we were able to categorize *Hevea* cultivars as primary, secondary, ternary, and quaternary clones. Primary clones were the individuals that were directly selected from the seedlings of Wickham germplasms and did not undergo artificial hybridization. They were regarded as the first generation. Secondary clones, regarded as the second generation were the individuals resulting from a cross of two primary clones. Ternary clones were the individuals derived from crosses of a secondary clone with either a primary or another secondary clone and considered as the third generation. Lastly, quaternary clones were the individuals whose parents were a ternary clone and either a primary, secondary, or another ternary clone and considered as the fourth generation. Two hundred and eight of the 335 *H. brasiliensis* accessions including 93 cultivated and 115 wild germplasms were planted in the field at 3 m spacing in row 7 m apart in 2008. The latex production was measured using a graduated cylinder after latex flow completely stopped. Each accession consisted of three individual trees. The measurement of latex production from three times of tapping were conducted for each individual tree. The others were preserved as epicormic shoots in the field after they were introduced in 1980s. In addition, 285 individuals from seven hybrid segregation populations were planted at 1.5 m apart in rows 3 m apart in fields at the Experimental Station of RRI-CATAS in 2015. The NLR of the multi-hybrid accessions was histologically evaluated[19]. 25 accessions with more secondary laticifer rings (ML) and 24 accessions with few secondary laticifer rings (FL), as well as five primary clones (PB86, Tian

Ren 31-45, Hong Shan 67-15, GT1, Guang Xi 6-68), were used for genotype identification by PCR. The leaves of all selected accessions were harvested for DNA extraction.

Plantlets of rubber tree clone CATAS7-33-97 (also known as 'Reyan7-33-97') were grown in a field at the experimental farm of the Chinese Academy of Tropical Agricultural Sciences (CATAS). They were used for experimental morphological study on secondary laticifer differentiation according to the method[15]. We selected 140 epicormic shoots for treatment at the stem of the second extension unit[15]. Of these shoots, fifty were treated with water as a negative control and another fifty shoots were treated with 20 μM COR (Sigma, USA) and 10 shoots were treated with mechanical wounding as positive controls. The treatment of interest was the 30 shoots treated with 10, 100, and 500 μM synthetic PSK (ChinaPeptides, China) for each of 10 shoots, respectively. Samples of bark tissue, including the cambial region, were collected from the stem treated with either COR or water for 1 h, 2 h, 1 d, and 3 d after their respective treatments before freezing them in liquid nitrogen for RNA extraction. The samples were collected from 10 shoots at each time interval. Bark samples, including xylem tissue, were collected from the rest of the shoots 10 days after treatments, and then they were fixed in 80% ethanol for cytological observation.

Ten-year-old virgin trees of CATAS7-33-97, CATAS8-79, PR107, and regularly tapped trees of CATAS7-33-97 and CATAS8-79 were grown in the field of the experimental farm of the CATAS. Latex samples from virgin trees were collected at the first, fifth and ninth tappings while latex samples from regularly tapped trees were collected two days after being tapped with an S/2 d3 tapping system. The latex samples from 10 trees were combined as one sample and collected on ice for RNA extraction. Three biological replications were conducted.

The self-incompatible wild-type TKS variety 'TK1151' was selected as the recipient plant of target genes from *H. brasiliensis*[10]. The TKS plants were cultivated at 21 °C with a 16/8 h photoperiod in the greenhouse of the Institute of Genetics and Developmental Biology, Chinese Academy of Sciences.

### Genome library construction and sequencing
The high-molecular-weight DNA of CATAS8-79 was extracted from leaf samples using the DNAsecure Plant Kit (Tiangen Biotech, China). The DNA was sheared to ~20-kb fragments. Following the manufacturer's protocols[41], PacBio-CLR (continuous long-read) libraries were constructed and sequenced on a PacBio Sequel sequencing platform (BGI, China). For genome size estimation and whole genome resequencing, the genomic DNA was extracted according to the manufacturer's procedures of the Plant Genomic DNA Kit (Tiangen Biotech, China). Using the NEBNext Ultra DNA Library Prep Kit (NEB, USA), 350-bp libraries were constructed and then sequenced using the BGISEQ-500 platform (BGI, China). Genome size estimation was performed based on the distribution of K-mer values obtained using Jellyfish (version 2.2.3)[42].

## Hi-C library construction and sequencing

Hi-C libraries were created and sequenced by Anoroad Company (China). Briefly, the leaves were fixed with formaldehyde and digested with HindIII/MboI enzymes overnight. The digested samples were labeled with biotin-14-dCTP and then blunt-end digested to prepare ends for ligation. After ligation, the biotin-labeled Hi-C samples were collected and amplified by PCR. The final paired-end sequencing libraries were sequenced using the Illumina HiSeq X Ten platform (BGI, China).

## BioNano genome map construction

Based on the manufacturer's procedures of the BioNano Prep Plant DNA isolation kit (BioNano Genomics, USA), high-molecular genomic DNA was extracted from young fresh leaves of CATAS8-79. The genomic DNA was purified by washing buffer, embedded in a low melting point agarose layer, digested with the BspQI enzyme, and labeled with the IrysPrep Labeling mix (BioNano Genomics, USA). The labeled DNA was loaded into IrysChips (BioNano Genomics, USA) and imaged on an Irys imaging instrument in BerryGenomics Company (China).

## De novo genome assembly and validation of CATAS8-79

To generate clean PacBio data, the raw reads generated from the PacBio platform were filtered using the SMRTAnalysis package (version 3.1) with default parameters. We obtained 192 Gb of high-quality PacBio clean data that were de novo assembled into contigs using CANU (version 1.7) with the following parameters: useGrid = true, minThreads = 4, genomeSize = 1300 m, minOverlapLength = 600, minReadLength = 1000. The genome assembly method HERA[43] was then used to extend the length of contigs based on the parameters as follows: InterIncluded_Side = 30000, InterIncluded_Identity = 99, InterIncluded_Coverage = 99, MinIdentity = 97, MinCoverage = 90, MinLength = 5000, MinIdentity_Overlap = 97, MinOverlap_Overlap = 1000, MaxOverhang_Overlap = 100, MinExtend_Overlap = 500.

To build the scaffolds, the raw BioNano data were filtered by removing the low-quality data. The optical map was further de novo assembled using IrysSolve (version 3.0) provided by BioNano Genomics. Hybrid scaffolds were assembled from the PacBio assembly and BioNano optical maps using the script hybridScaffold.pl with the following parameter: -B 1 -N 2. To assemble pseudochromosomes, the Hi-C read pairs were filtered using default parameters. Using Juicer[44], the cleaned Hi-C data were aligned to the hybrid scaffolds. Based on the alignments, the hybrid scaffolds were further assembled into 18 pseudochromosomes by 3D-DNA pipeline[45]. Finally, the assembled sequences were then filtered using Purge_Haplotigs[46] and Redundans[47] as default parameters. The NGS reads were used to polish the assembly. All clean reads were mapped back to the assembly using Pilon (version 1.22) based on the default parameters. The quality of the assembled genome was evaluated by the following approaches: (1) we mapped the NGS data to the assembly using BWA (version 0.7.17-r1188); (2) we assessed the continuity and completeness of assembly by BUSCO (version 4.1.4), LAI[48], and Merqury[49].

## RNA extraction, transcriptome sequencing, and Iso-Seq

Total RNA was extracted from each tissue using the RNAprep Pure Plant Kit (Tiangen Biotech, China), and the genomic DNA was removed using RNase-Free DNase I (Takara, Japan). The integrity and quality of RNA were evaluated on a 1.5% agarose gel and NanoDrop 2000 (Thermo Fisher Scientific, USA), respectively. For transcriptome sequencing, total RNA of each tissue was separately constructed into paired-end cDNA libraries using the NEBNext Ultra RNA library Prep Kit (NEB, USA). The seven libraries were respectively sequenced on the BGISEQ-500 platform. The read counts of matched genes were obtained by mapping the clean reads to the CATAS8-79 genome. The expression levels of matched genes were derived and normalized to FPKM (Fragments Per Kilobase of exon per Million fragments mapped) using Trinity (version 2.4.0)[50]. For Iso-Seq, equivalent amounts of RNA from each of the seven tissues were mixed, and cDNAs were synthesized using the SMARTer PCR cDNA Synthesis Kit (TaKaRa, Japan). The Iso-Seq libraries were constructed following the SMRT bell manufacturer's procedures and sequenced on the PacBio RS II platform (BGI, China).

## Genome annotation

The repetitive sequences were annotated by both homology-based and de novo approaches. For homolog repeat annotation, the CATAS8-79 genome was aligned with RepBase 21.01 by RepeatMasker (version 4.0.6)[51]. For de novo annotation, LTR_FINDER (version 1.06)[52] was used to generate a de novo repeat library, and the library was further detected by RepeatMasker (version 4.0.6) according to the default parameters. Protein-coding genes were predicted using three approaches: protein homolog, transcripts assembly, and ab initio predictions. Firstly, H. brasiliensis protein sequences from the Gen-Bank of NCBI were aligned to the CATAS8-79 genome using BLAST with an e-value <10⁻⁵. Secondly, de novo and genome-guided assemblies from RNA-seq reads of the seven tissues of interest were obtained using Trinity (version 2.4.0)[50]. The transcripts constructed from RNA-seq and Iso-seq data were aligned to the CATAS8-79 genome using the BLAST-Like Alignment Tool. Thirdly, ab initio gene predictions were performed using four sorts of softwares: AUGUSTUS[53] (version 3.2.3), GeneScan[54] (version 1.0), GlimmerHMM[55] (version 3.04), and SNAP[56] (https://github.com/KorfLab/SNAP). All predicted gene structure evidence was finally integrated with EVidenceModeler[57] (version 1.1.1). The final high-confidence gene model set was constructed by removing the incomplete gene models and those with an internal stop codon, or an FPKM of 0 across the RNA-Seq/Iso-seq data. The constructed high-confidence genes were functionally annotated according to the proteins deposited in GO, KEGG, and Swiss-Prot with an e-value <10⁻⁵.

## Gene family and synteny analysis

To identify gene families, we analyzed protein-coding genes from 12 plant species, including Hevea brasiliensis Muell. Arg., Manihot esculenta Crantz, Ricinus communis L., Populus trichocarpa Torr. & Gray, Carica papaya L., Arabidopsis thaliana (L.) Heynh., Camellia sinensis (L.) O. Ktze., Medicago truncatula L., Fragaria vesca L., Vitis vinifera L., Taraxacum kok-saghyz Rodin, and Oryza sativa L. Using BLASTP with an e-value <10⁻⁵, the homologous genes were identified by all-against-all comparison, and then they were clustered with Orthofinder (version 2.3.3)[58] to identified gene families. The single-copy orthologous gene families were aligned by MUSCLE (version 3.8.31)[59] and used to construct a phylogenetic tree by RAxML (version 8.2.12)[60] with a JTT + I + G model and 1000 bootstrap replicates. The divergence time among 12 species were estimated by the MCMCTree program of PAML (version 4.5)[61] and corrected by Timetree (http://www.timetree.org/). The expansion and contraction of gene families were calculated using CAFÉ[62]. The transcriptional factor families were identified by mapping to the Plant Transcription Factor Database (http://itak.feilab.net/cgi-bin/itak/index.cgi) by iTAK with default parameters[63]. All-against-all BLASTP searches with an e-value <10⁻⁵ were performed to identify syntenic gene blocks in intra-genome (CATAS8-79 and GT1) and inter-genome (CATAS8-79 vs GT1; CATAS8-79 vs cassava). At least five synteny gene pairs were defined as one syntenic block based on the MCScanX package with default settings. The adjacent blocks were then merged into large syntenic blocks. The size of large syntenic blocks over 10 Mb was selected for further analysis.

## Mapping of genome reads and variation calling

The high-quality reads were aligned against the CATAS8-79 assembly using BWA (version 0.7.17-r1188) with the following parameter: mem -t 4 -M −R. The BAM alignment files were subsequently filtered and

sorted by SAMtools (version 1.3)[64] using the following parameter: samtools view -q30. 385. VCF files of each accession were consolidated into a single VCF file for SNP calling using GATK (version 3.4)[65] based on the following parameters: --clusterWindowSize 10 --filterExpression "MQ0 >= 4 && ((MQ0 / (1.0 * DP)) > 0.1)" --filterName "HARD_TO_VALIDATE"–filterExpression "DP < 5" --filterName "LowCoverage" --filterExpression "QUAL < 30.0" --filterName "VeryLowQual" --filterExpression "QUAL > 30.0 && QUAL < 50.0" --filterName "LowQual" --filterExpression "QD < 1.5" --filterName "LowQD". The subsequent SNPs were further filtered with the following parameters: maf ≥ 0.05 and missing rate ≤10%. The final SNPs were annotated using snpEff (version 4.3).

### Population genetic analyses

For a phylogenetic tree analysis 336,200 SNPs located in the coding sequence (CDS) region in 336 *Hevea* accessions were used. Using RAxML (version 8.2.12), we constructed a phylogenetic tree with 100 bootstrap replicates. The same dataset was also used for population structure analysis. The population structure of the rubber tree accessions was determined using fastStructure (version 1.0)[66] with all SNPs at four-fold degenerated sites for each K value ranging from 2 to 10 without the individuals located far from the major clades in the phylogenetic tree and the cultivars with the rubber tree primary clone Tjir1's or PR107's pedigree. In addition, the PCA was performed using SNPRelate (version 1.16.0)[67]. Three-dimensional coordinates were plotted for the 336 *Hevea* accessions. The $F_{st}$ and π values were calculated by VCFtools (version 0.1.15)[68] while using a 100-kb window with a step size of 100 kb for each sub-population. LD coefficients (r2) between pairwise SNPs were calculated using Plink (version 1.90b5)[69] and the parameters: --ld-window-r2 0 --ld-window 99999 --ld-window-kb 1000. The results were used to estimate LD decay. To identify genomic regions affected by artificial selection, we performed $F_{st}$, π and XP-CLR (version 1.0) analyses on Cul and WMG based on a sliding window of 20 kb and a step size of 20 kb.

### Phenotyping and genome-wide association study

We used 208 10-year-old *Hevea* accessions (in triplicate) including 93 cultivated and 115 wild germplasms for phenotyping (Supplementary Data 7). The bark of individual trees was fixed in 80% ethanol overnight and then treated with iodine and bromine in glacial acetic acid before embedding in paraffin after ethanol dehydration[15]. Sections were cut with a microtome and stained with fast green. The NLR within the area between the cambia and innermost layer of stone cells was counted under a Leica DMLB microscope (Leica, Germany). The GWAS was conducted based on the linear mixed-model algorithm implemented in the EMMAX program. A kinship matrix generated with the EMMAX was used to correct the population stratification. A modified Bonferroni correction was used to determine the genome-wide significance thresholds of the GWAS corresponding to raw P values of $-\log_{10}P > 4$.

### RNA isolation, cDNA synthesis, semi-quantitative, and qRT-PCR analyses

Total RNA was extracted using the RNAprep Pure Plant Plus Kit (Tiangen Biotech, China). The integrity and quality of total RNA were evaluated in 1.5% agarose gel and by the NanoDrop 2000 (Thermo Fisher Scientific, USA), respectively. Approximately 1 μg of total RNA was used to synthesize cDNA using a RevertAid™ First Strand cDNA Synthesis Kit (Thermo Fisher Scientific, USA). Semi-quantitative PCR was performed as follows: 96 °C for 5 min, followed by 30 cycles (for target gene *HbACS4*) or 23 cycles (for reference gene *HbUBC3*) of 96 °C for 30 s, 60 °C for 30 s, and 72 °C for 30 s, and a final extension at 72 °C for 10 min. The amplified products were separated on 2% agarose gels. qRT-PCR was performed using the CFX384 real-time PCR system (Bio-Rad, USA) with the SYBR Prime Script RT-PCR Kit (TaKaRa, Japan). Relative gene expression levels were respectively normalized by the expression levels of *18S-rRNA* (for latex samples from *Hevea* virgin trees), *HbUBC2b* (for latex samples from the tapped trees of CATAS8-79 and PR107), *HbUBC3* (for bark samples from the stem of epicormic shoots of CATAS7−33-97), and *TkGAPDH* (for samples from TKS). Three biological replicates were conducted. All primers were listed in Supplementary Data 18.

### Yeast one-hybrid assay

The promoter region of *HbPSK5* was cloned into the pAbAi vector (Clontech, USA), generating pHbPSK5-AbAi. The CDS of 29 *HbMYCs* were cloned into the pGADT7 vector. The pGADT7-HbMYCs and pHbPSK5-AbAi vector were co-transformed into the yeast strain Y1HGold. pGADT7-Rec-p53 and p53-AbAi were used as positive controls (CK+) while pGADT7 and pHbPSK-AbAi were used as negative controls (CK-). Transformants were grown on SD/−Leu/AbA medium with 200 ng/mL Aureobasidin A (AbA) for 3 d at 30 °C.

### Transient dual-luciferase assays

Each of the CDS of five *HbMYCs* (-10, -15, -18, -25, -26) was individually cloned into the pCAMBIA2301 effector vector. To generate a reporter construct, the promoter region of *HbPSK5* was cloned into the pGreenII 0800-LUC vector. All the recombinant vectors were confirmed by sequencing. The reporter vector and nine effector vectors were separately transferred into *A. tumefaciens* (strain GV3101) and cultured overnight in LB liquid medium with 50 mg/L kanamycin. The overnight cultures were collected by centrifugation, adjusted to an $OD_{600} = 0.2$ with the filtration buffer (10 mM 2-(N-Morpholino) ethanesulfonic acid (MES, Wuhan BioRun Biosciences Co., Ltd.), 150 mM acetosyringone, and 10 mM $MgCl_2$) and then incubated for 4 h at room temperature. The effecter and reporter suspensions were mixed equivalently and carefully press-infiltrated onto healthy leaves of 6-week-old *Nicotiana benthamiana*. After 3 days of infiltration, the infected discs were harvested for total protein extraction. The LUC and REN luciferase activity was detected using a Dual-Luciferase Reporter Assay System (Promega, USA) following the manufacturer's instructions. The LUC value was normalized to that of REN. Sixteen independent biological replicates were performed for each combination.

### Electrophoretic mobility shift assay

The EMSA was performed with the Electrophoretic Mobility Shift Assay kit (Invitrogen, USA) following the manufacturer's instruction. The CDS of *HbMYC18* was cloned into the prokaryotic expression vector pET-28a, and then transformed into *Escherichia coli* strain BL21 (DE3) after sequencing verification. The recombinant HbMYC18-His protein was expressed and affinity-purified using $Ni^+$ affinity resin (GE, USA). The 1.2-kb promoter region of *HbPSK5* was amplified from the CATAS8-79 genome. The binding reaction was performed by incubating double-stranded DNA of the *HbPSK5* promoter with different concentrations of purified recombinant HbMYC18-His protein at room temperature for 30 min in a total volume of 15 μL. The reaction system contained 20 mM Tris-HCl (pH 7.6), 30 mM KCl, 0.2% (w/v) Tween-20, 1 mM DTT, and 10 mM $(NH_4)_2SO_4$. The DNA-protein complexes were separated on 5% polyacrylamide gels and captured on an Azure Biosystems c300 imaging system (Azure, USA).

### Functional identification of *HbMYC26* and *HbPSK5* in *Taraxacum kok-saghyz* Rodin

To construct the plant transformation plasmids *35S::HbMYC26* and *35S::HbPSK5*, the cDNA fragment of *HbMYC26* and *HbPSK5* was cloned into the pFGC5941 binary vector (www.chromDB.org). The constructed plasmids were introduced into *Agrobacterium tumefaciens* strain AGL1, and TK1151 plants were transformed based on leaf disc method[70]. The phenotypes of $T_0$ transgenic lines were examined after being propagated by tissue culture. Fresh root samples from 3-month-old plants of *35S::HbMYC26* and *35S::HbPSK5* were fixed in 80% (v/v)

ethanol for 24 h and then sliced by a vibratome (Leica, Germany). The vibratome parameters were as follows: speed = 1.00 mm/s, amplitude = 1.0 mm, and thickness = 100 μm. Sections were stained with Oil Red O (Sigma, USA) and mounted in 60% (v/v) glycerol[10]. Pictures were taken with a light microscope SZX16 (Olympus, Japan). The laticifer cell index was used to estimate the abundance of laticifer cells in tissues and was calculated as the total area of the Oil Red O stained laticifer cells divided by total area of the whole root section using ImageJ software. The NR of transgenic or mock lines was extracted and determined as follows[10]. 100 mg root powder was mixed with 1 mL toluene for 2 h at 50 °C, and then centrifuged 13,800 × g for 10 min at room temperature. The upper organic phase was transferred to a new tube and added 2 volumes of methanol to precipitate natural rubber. The rubber samples were dried and redissolved in 1 mL of toluene and measured using Fourier Transform Infrared (FTIR) spectroscopy (Thermo Scientific, Waltham, MA, USA).

## Statistical analysis and reproducibility
The statistical analysis was performed using Student's *t*-test (two-sided). The semi-quantitative PCR experiment and EMSA experiment in this study were performed independently two times, and other experiments in this study were performed independently at least three times.

## Reporting summary
Further information on research design is available in the Nature Portfolio Reporting Summary linked to this article.

## Data availability
The genome assembly and annotation generated in this study have been uploaded to the open dissemination research data repository Zenodo with a website: https://doi.org/10.5281/zenodo.7123623. The raw sequencing data and transcriptome sequencing data generated in this study have been deposited to the National Genomics Data Center (NGDC, https://ngdc.cncb.ac.cn/) under project number PRJCA004986. Published transcriptome sequencing data (Gene Expression Omnibus (GEO) accession number GSE80596) is used for expression analysis in the present study. Source data are provided with this paper.

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

## Acknowledgements

The research was supported by the National Key Research and Development Program of China (2018YFD1000502 to W.M.T.), the Earmarked Fund for China Agriculture Research System (CARS-33-YZ1 to W.M.T.), the National Natural Science Foundation of China (31700601 to J.C., 31870590 to W.M.T., 31800577 to S.Z., 31800578 to X.D.), the Central Public-interest Scientific Institution Basal Research Fund for Chinese Academy of Tropical Agricultural Sciences (1630022020021 to J.C., 1630022021001 to S.Z.), the Hainan Provincial Science and Technology Special Fund (ZDYF2022XDNY252 to J.C.) and the Hainan Provincial Natural Science Foundation of China (320CXTD442 to M.S.).

## Author contributions

W.M.T., C.L., and J.L. designed and supervised the overall research. J.X. constructed the BioNano genome map. Q.G., H.D., J.Z., Z.W., and Y.L. assembled and annotated the CATAS8-79 genome. J.C., S.Y., Yan Li, X.D., X.H., X.G., X.Z., Y.H., X.Ze., W.L, S.W., M.S., and S.Z. collected rubber tree accessions and conducted phenotypic analysis. Q.G., B.G.,

and W.X. performed the population analysis and GWAS. J.C. and W.M.T. performed experimental morphological study. J.C., S.Y., S.W., S.Z., and D.G. performed gene expression analysis. X.X., X.F., and C.H. conducted TKS transgenic and histochemical analysis. J.C., L.K., X.W., W.M.T., C.L. L.Z., S.P., J.W., B.H., H.Y., and J.L. analyzed data and wrote the paper with input from all other authors.

## Competing interests

The authors declare no competing interests.

## Additional information

[1]National Key Laboratory for Tropical Crop Breeding, Chinese Academy of Tropical Agricultural Sciences, Sanya 572024, China. [2]Ministry of Agriculture and Rural Affairs Key Laboratory of Biology and Genetic Resources of Rubber Tree, Rubber Research Institute, Chinese Academy of Tropical Agricultural Sciences, Haikou 571101, China. [3]State Key Laboratory Breeding Base of Cultivation and Physiology for Tropical Crops, Rubber Research Institute, Chinese Academy of Tropical Agricultural Sciences, Haikou 571101, China. [4]State Key Laboratory of Plant Genomics, Institute of Genetics and Developmental Biology, Innovation Academy for Seed Design, Chinese Academy of Sciences, Beijing 100101, China. [5]Genomics and Genetic Engineering Laboratory of Ornamental Plants, College of Agriculture and Biotechnology, Zhejiang University, Hangzhou 310058, China. [6]Qi Biodesign, Life Science Park, Beijing 100101, China. [7]School of Life Sciences, Institute of Life Sciences and Green Development, Hebei University, Baoding 071002, China. [8]University of Chinese Academy of Sciences, Beijing 100049, China. [9]Key Laboratory of Animal Ecology and Conservation Biology, Institute of Zoology, Chinese Academy of Sciences, Beijing 100101, China. [10]Ministry of Agriculture and Rural Affairs Key Laboratory of Tropical Crop Biotechnology, Institute of Tropical Bioscience and Biotechnology, Chinese Academy of Tropical Agricultural Sciences, Haikou 571101, China. [11]BGI Genomics, BGI-Shenzhen, Shenzhen 518083, China. [12]Institute of Chinese Materia Medica, China Academy of Chinese Medical Sciences, Beijing 100700, China. [13]Key Laboratory of Plant Resources Conservation and Germplasm Innovation in Mountainous Region, Ministry of Education, Institute of Agro-Bioengineering, College of Life Sciences, Guizhou University, Guiyang 550025, China. [14]Hainan Yazhou Bay Seed Laboratory, Sanya 572024, China. [15]These authors contributed equally: Jinquan Chao, Shaohua Wu, Minjing Shi, Xia Xu, Qiang Gao, Huilong Du. ✉e-mail: jyli@genetics.ac.cn; cliang@genetics.ac.cn; wmtian@163.com

