## [Peer Review File · Nature Communications]

Genomic insight into domestication of rubber treeReviewers' Comments:

Reviewer #1:

Remarks to the Author:

The authors of the manuscript investigate genomic footprints of domestication in the rubber tree (*Hevea brasiliensis*) combining a genome assembly, and 335 resequenced accessions of wild and cultivated rubber trees. Understanding processes of domestication in trees is a major goal in evolutionary biology, and this manuscript makes one step forward in that direction. In particular, the rubber tree has not been studied yet, and this manuscript should provide insights into genes related to the number of laticifer rings selected during rubber breeding (domestication? See my remark below). The sampling is comprehensive, and a special effort is made to obtain wild genotypes, which is crucial to understand the domestication process and also assess future sources of alleles for breeding programs. However, 1) some parts of the ms are not clearly explained, 2) the ms is in general too species-centered, 3) population structure should be better presented, and 4) selection tests should take into account admixed individuals (same for GWA analyses) and be done using additional statistics. I have an additional remark concerning the term "domestication" for the rubber tree. Please see my comments below.

Major comments

- The manuscript is too rubber-tree centered, a first paragraph for non-specialist would be appreciated before starting introducing the rubber tree: why is it interesting to understand domestication in trees and understanding the associated genomic bases? In particular in perennials, see:
<https://par.nsf.gov/servlets/purl/10025433>
<https://pubmed.ncbi.nlm.nih.gov/24290193/>
<https://nph.onlinelibrary.wiley.com/doi/10.1111/j.1469-8137.2012.04253.x>
- The introduction well describes the traits of interest studied here, but there is a lack of information on what we know about the progenitor(s) of the cultivated rubber tree. The authors sampled different populations in South America. The choice of those populations should be justified in the Introduction, at least briefly. At the moment, this information only comes in the results part, and it is not straightforward to understand.
- The last paragraph in the introduction is too vague, the authors should clearly state the questions they will answer, e.g. 1) What is the genome structure? 2) Gene gain and loss during domestication or diversification? 3) Relationships among wild and cultivated rubber trees? 4) Footprints of selection in the genomes etc... I discovered those objectives along the results part, which makes the results part challenging to understand.
- The population structure analyses should be presented in the main text, at least the most likely K value (and the deltaK statistic should be provided in supp mat); the PCA is also interesting. In addition, the STRUCTURE barplot for $K > 4$ should be provided in supplementary material to check whether for $K > 4$ there is no additional substructure observed.
- F_{ST} and π can be computed only for panmictic populations without admixed samples: as F_{ST} and π rely on the estimate of neutral site frequency spectrum; the occurrence of admixed individuals skew the SFS and then the selection inferences. The authors must therefore remove the admixed individuals and define the population of "pure" individuals = group of individuals assigned with a membership coefficient $> X\%$ to a given cluster.
- In addition, π and F_{ST} have their own limits, and the authors must use, at least, an additional statistic. For instance, the composite likelihood ratio (CLR) statistic and/or the μ statistics (<https://github.com/alachins/raisd>).
- GWAS should also be run without the admixed individuals.
- The authors talk about "domestication" of rubber trees that started 150 years ago. Can we talk about domestication in the case of the rubber tree, or only of breeding? A discussion on that part may be appreciated.

Minor comments:

- Line 38: precise genome size
- Line 38: "upgraded" here, but in the main text, it sounds like it is a new genome? Please rephrase, correct
- Line 44: PSK, please define for a non-specialist.
- Line 46: Taraxacum kok-saghyz, unclear for non-specialist, explain.
- Line 50: One paragraph which introduces more broadly the questions (see major comments) would be a plus.
- Line 67: 150 years, 5 generations then, even less when you take in account clonal propagation? Add this information?
- Line 69: What do you mean by "genuine clonally propagated crop"? please rephrase.
- Lines 97-100: In the abstract you say "upgraded", but here it sounds like it is a new genome? Please rephrase, correct either here or in the abstract.
- Line 102: correct 1,583Mb, by 1.53 Gb (to make the parallel with line 99).
- Line 112: "repeats", precise, is it only transposable elements (TEs), or also SSR or? And what is the proportion of TEs in the genome? This information should be added in Table 1.
- Line 118: unclear what does "CATAS" mean without any information before. Please define.
- Line 152: the authors should present the STRUCTURE outputs in the main text and remove the admixed individuals for selection analyses (see major comments).
- Line 160: "decrease of genetic diversity ... food crop". That depends, see Gaut et al 2015 and Cornille et al 2014 (reference above), actually you do not expect that in trees. Please rephrase.
- line 165: "cause by fewer occurrence of sexual recombination". It sounds counterintuitive, you should expect that few recombinations lead to lower diversity... Please rephrase.
- line 196: and why not using as well sweep regions obtained with pi?
- line 222: which genotypes did you use?
- Figures : all figures should be self-explanatory, i.e. with species names, number of samples used, and if needed, method used.
- Figure 1: add species names, number of accessions (wild and crop? How many?) and any relevant data used to make the figure (assembly, genome size, n50...).
- Figure 1: (VI) gene expression level for different organs, all organs pooled? Precise.
- Figure 1: (IV) Why did the authors present copia of only retrotransposon and gypsy? Why not showing all the TEs?
- Figure 4 explains a lot of information. Can't it be split? In addition the number of samples used is unclear. For (e) 208 cultivated and wild germplasm? and (f) 127 cultivars and 208 wild germplasm. Please make it clearer.

Reviewer #2:

Remarks to the Author:

I have read the manuscript by Chao, Wu, Shi, Xu, Gao, Du et al. entitled "Genomic insight into domestication of rubber tree" with great interest. Based on a combination of long-read and long-range technologies, the authors describe a high-quality assembly of the CATAS 8-79 rubber tree genome. The genome assembly is highly contiguous and chromosome-scale and represents a slight improvement over the current reference genome (GT1 individual). In addition, they sequenced more than 300 accessions and performed association studies focusing on key domestication genes. In particular a major trait, the number of laticifer rings, related to rubber content.

First, I was bothered by the fact that the generated data (raw reads, resequenced accessions, RNA-Seq data) is not available. Also, genome assembly and annotation could not be downloaded. Therefore, I could not verify the quality of the scientific work. Authors must share the data with the editor and reviewers and ensure that the data will be available when the article is published.

More generally, the article contains frequent shortcuts and lacks explanations which complicate

understanding and raise doubts on the methods used by the authors.

Several years ago, two important loci related to latex production (LOL, Lot Of Latex) were identified (<https://doi.org/10.1111/nph.15253>). The authors also established a genetic link between jasmonic acid (JA) signaling and latex production in laticifers. I wondered if these genomic regions and associated genes had homologs in rubber-producing plants.

The authors annotated several PSK genes in the Hv genome, one would expect to have a comparative analysis that could trace the evolutionary history of this important gene family.

- The abstract contains undefined abbreviations (MYC, JA, PSK), repetitions and unclear statements (by promoting laticifer formation in *Taraxacum kok-saghyz*).
- L100. The genome size was estimated by the kmer distribution but Supplementary Table 1 does not contain any size estimate. According to the methods section, the authors use Genomescope, but they must provide distribution plots and results.
- L100. Please describe the type of PACBIO data that was generated.
- L104. Supplementary Figure 1 should be a table.
- I wonder what is the heterozygosity rate of the rubber tree genome. I suspect that this rate is higher than 1%. The authors use the Pilon polisher which has been described to perform poorly in the case of heterozygous genomes and can lead to fragmented genes (<https://doi.org/10.1093/nargab/lqab034>).
- Also, the BUSCO score is higher for genome assembly than for gene models, and the difference is due to a higher proportion of fragmented genes, which is linked to the point raised earlier. Moreover, the proportion of duplicated genes is high, does it mean that the final assembly still contains allelic duplications? The authors should provide more information to the reader.
- L116. High-confidence gene models, how confidence was assessed?
- L119-120, the link between the number of syntenic blocks and higher contiguity is unclear.
- L127. I assume Hb stands for *Hevea brasiliensis*, but please add that information.
- L132. Clade A, please add information on existing clades.
- L145. Supplementary Table 8 contains a column titled "coverage". I assume the value represents the fraction of the reference genome that is covered. This coverage value is often close to 70-80%. How do the authors explain these low values?
- L189-190. "The expression patterns of the selected genes were associated with yield productivity". Authors need to go further and perform correlation analysis if they want to associate expression patterns with traits.
- L335. Please provide optical maps information, Supplementary Table S2 contains raw data metrics only.
- I could not find any information on how the RPKMs in Supplementary Table S6 were calculated.
- I suggest adding information about the current rubber tree reference genome in Table 1 along with the BUSCO and Merqury scores for both assemblies.
- Figure 1: a track is missing
- Figure 2c : "cycles" should be replaced with "nodes", and "between the cycles" with "edges".
- Figure 2d. How was the difference in latex production measured?
- Figure 4a. The Manhattan plot shows a significant signal on chromosome 15, as noted by the authors. However, they did not mention the signal on chromosome 13.
- Figure 4c. The scale is required. Also the figure legend mentions FPKM values, but Table S6 contains RPKM.

Reviewer #3:

Remarks to the Author:

This study is a team effort of several labs studying domestication of *Hevea brasiliensis* (rubber tree). Rubber tree is an important crop that only recently (~150 years) has been domesticated through breeding efforts. Due to the long generation time, only very few selection rounds by breeders were

possible, yet in current cultivars rubber production is several folds increased over wild populations. The study is a tour-de-force comprising of an improved genome assembly, population genomic analyses of more than 300 (cultivated and wild) accessions that led to the identification of a putative domestication gene (PSK5), its regulation by MYC transcription factors, and the functional verification of said gene's function on the physiological level (i.e. promoting the generation of laticifer cells from the vascular cambium). The study identifies a SNP upstream of the PSK5 gene, and a particular genotype of this SNP (GG) strongly correlates with high rubber production, not just across accession, but also in hybrid segregation populations. This certainly impressive work. Since I am not in a good position to judge the technicalities of the genomics analyses, I focus in my review on the identification of PSK5 as promoting laticifer cells. PSK signalling is a somewhat enigmatic signalling pathways as its functions seem to be diverse and context-dependent and only very few signalling components and transcriptional targets downstream of the PSK peptides and their cognate PSKR receptors are known. Even though PSK has been implicated in vascular cell fate control before, its proposed function in promoting laticifer cell fate is certainly novel and very interesting. However, the data presented here do at present insufficiently support a role for PSK in rubber production

Specific points:

-The correlation of genotype and rubber production is high and convincing but it is insufficiently explained how PSK5 might affect rubber production. The associated SNP itself is 700 bp upstream of the transcriptional start site. It is not clear whether the authors assume this also is the causative mutation. The manuscript is very vague on this point. Would associated indels be detected by their analyses? Were PSK5 alleles from different accession sequenced? If the SNP is indeed supposed to be causative, the plausible scenario is that PSK5 expression differs between genotypes. However, this is not experimentally verified by, e.g. by qPCR analyses in a few accessions. A mutation like this is a priori rather unlikely to have a large scale effect and PSK5 expression in the elite cultivar CATAS 8-79 is rather low compared to the other PSKs (possibly due to tissue specificity), therefore more efforts to strengthen the causal link are warranted.

-Related with the previous point, the transgenic TKS lines are overexpressors of the MYC transcription factor or the PSK target. Ideally one would compare the different alleles but obviously the upstream regulatory sequences from rubber tree can't be expected to yield insight in the heterologous system. However, while this limitation is acknowledged, it places even higher emphasis on the need for correlative PSK5/MYC expression analysis in accessions with different genotypes/rubber production

-Another concern with the transgenic lines is whether WT is the appropriate control. As far as I understand, TKS is transformed via leaf disc regeneration. It is not clear from the methods section whether these are regenerated or seed grown plants. Even if these are seed-grown, an empty vector-transformed transgenic line would be more appropriate than the WT. As regeneration manifests epigenetic changes and is mutagenic, a mock transformed line would rule out that increased laticifer cell production is unrelated to PSK/MYC expression. A line that only weakly expresses the respective transgene might also help here if it strengthens the correlation between laticifer cells and transgenes, as all the presented transgenic lines seem to show rather similar expression levels, which do not correlate with the phenotype.

-Fig4I should be quantified

-100 μ M of PSK seems rather high as a treatment, do lower doses show an effect as well? It is difficult to control how much PSK arrives at the site of action (presumably the cambium), my concern here is whether laticifer cell production might be a general stress response, somewhat in line with the observed increase by application of JA or by wounding. Lower doses would alleviate this concern, but I fully acknowledge that a requirement for a high PSK dose does not imply a problem with the conclusion per se.

-In line 124/125 a citation seems to be missing.

-Finally, a possibly naïve question from a GWAS non-expert: Given the few rounds of recombination during domestication, is it not very surprising to see that many significant loci in the Manhattan plot, suggesting many low impact variants?

Reviewer #4:

Remarks to the Author:

The work reported in this manuscript represents an extremely important and welcome addition to the genomic information available on the rubber genome. The Hevea genome information in the public domain has lagged behind comparable information for most important crops and it is good to see this addressed with a data set based on relatively current sequencing and assembly technologies. The inclusion of both cultivated clonal material and material from wild germplasm collections is also welcome. The comparison with other genomes particular that of Cassava with which Hevea shares a recent evolutionary history is also valuable and will help greatly in leveraging information from other species to support functional annotation.

Though I am happy that this represents a major contribution to Hevea genetics and biology I have some concern about the contrasts that have been made between the wild and cultivated genepools. This is for two reasons - first of all because of the area of the original Wickham collections is not represented in the wild component of the comparison and second because the cultivated germplasm is, inevitably for Hevea, heavily biased by over representation of several genotypes in the pedigrees. e.g., RRIM600 and PR107. However, though I believe that this does not negate the conclusions the authors make I would suggest that they need to acknowledge this in their conclusions. On this basis the identification of HbPSK5 as a key domestication gene might have been somewhat suspect. However, the functional analysis through transformation of *Taraxacum* goes a considerable way towards overcoming any such objection and I am happy to accept their argument.

On a technical point, I find many of the diagrams are too small to read. While I realise that there are severe constraints in terms of journal formats, at least for supplementary figures I see no reason for such small diagrams.

While I acknowledge the deposition of the sequence data and I can see that accessions have been created I am not able to access them as they are set to be released on the 1st January next year. I cannot therefore identify what in fact will be made public. I would expect genomic and transcriptomic sequence data, assembly data and annotated genes but it is not possible to verify from the text or online that this is indeed the case. I would also hope that the intention is to make the data available in a more tractable form for many researchers by making available in a browser format and would welcome a statement to that effect in the manuscript.

With these reservations I am happy to recommend publication of this manuscript.

Point-by-point responses to the Reviewers' comments on the MS
(NCOMMS-22-20274)

We here gratefully thank all the reviewers for their critical comments. The manuscript is now dramatically improved, which we hope meet with approval. The followings are the point-by-point responses to the reviewers' comments on the MS (NCOMMS-22-20274).

Point-by-point responses to the Reviewer1's comments

Comment 1: “The manuscript is too rubber-tree centered, a first paragraph for non-specialist would be appreciated before starting introducing the rubber tree: why is it interesting to understand domestication in trees and understanding the associated genomic bases? In particular in perennials see:

<https://par.nsf.gov/servlets/purl/10025433>

<https://pubmed.ncbi.nlm.nih.gov/24290193/>

<https://nph.onlinelibrary.wiley.com/doi/10.1111/j.1469-8137.2012.04253.x>”

Response 1: Thank you for your valuable suggestions. We have revised accordingly and added a paragraph as the followings: “Plant domestication is a genetically dynamic and ongoing process of a wild species to adapt to human use by selecting specific traits¹⁻². Knowledge about the genetic basis of plant domestication mainly comes from annual crops³⁻⁴. It appears that the domestication process of long-lived perennials differs dramatically from that of annuals in that perennials tend to be domesticated at a slower rate, are typically out-crossing, have extended juvenile periods, are propagated clonally, and exhibit significantly fewer domestication syndrome traits¹⁻². The dynamics of domestication in perennials differs markedly from that of annuals, and the genetic basis of domestication in perennials has been studied in less depth².”

Comment 2: “The introduction well describes the traits of interest studied here, but

there is a lack of information on what we know about the progenitor(s) of the cultivated rubber tree. The authors sampled different populations in South America. The choice of those populations should be justified in the Introduction, at least briefly. At the moment, this information only comes in the results part, and it is not straightforward to understand.”

Response 2: Thank you for your kindly suggestions. We have revised and provided related information in Introduction section as the followings: “The rubber tree was primarily wild-harvested in South America before Henry Wickham collected rubber tree seeds in Boim of Brazil and introduced them into Kew Botanic Garden of the United Kingdom, starting the domestication process of rubber tree in 1876^(Ref. 12). So far most of the *Hevea* cultivars around the world are domesticated from Wickham collections. To widen the genetic basis for generating new clones with regarding to productivity, disease resistance, and tolerance to many environmental conditions, the International Rubber Research and Development Board (IRRDB) organized experts to collect wild Amazonian rubber tree germplasms from several districts of the states of Acre, Rondonia, and Mato Grosso of Brazil in 1981, termed as 1981’ IRRDB wild germplasms⁵”.

Comment 3: “The last paragraph in the introduction is too vague, the authors should clearly state the questions they will answer, e.g. 1) What is the genome structure? 2) Gene gain and loss during domestication or diversification? 3) Relationships among wild and cultivated rubber trees? 4) Footprints of selection in the genomes etc... I discovered those objectives along the results part, which makes the results part challenging to understand.”

Response 3: Thank you for your valuable suggestions. The last paragraph in the introduction has been revised accordingly as the followings: “Understanding the genetic basis of rubber tree domestication is crucial for further improving natural rubber production to meet its increasing demand worldwide. For this purpose, we here assembled a high-quality genome of the rubber tree clone CATAS8-79, resequenced 335 accessions and performed a population genetic analysis and genome wide

association study (GWAS) based on the map of genome variations and a major domesticated trait NLR. The revealed domestication-related molecular signals and one identified key domestication-related gene will be valuable for improving rubber tree and accelerating the domestication of other rubber-producing plants and possibly, perennial crops.”.

Comment 4: “The population structure analyses should be presented in the main text, at least the most likely K value (and the deltaK statistic should be provided in supplement); the PCA is also interesting. In addition, the STRUCTURE barplot for $K > 4$ should be provided in supplementary material to check whether for $K > 4$ there is no additional substructure observed.”

Response 4: Thanks for your suggestions. We have presented the population structure and the PCA in the main text in the revised Fig.2. As suggested, the STRUCTURE barplot for $K = 2$ to 10 is provided in supplementary Fig. 4. It is noteworthy that an additional group appears in cluster cultivars for $K > 4$. This group mainly consists of the pedigree of rubber tree primary clone PR107. Furthermore, the source of the four populations is clear. These data showed that the $K=4$ for four populations is reasonable.

Fig. 2 Geographic distribution and population analysis of 336 *Hevea* accessions. (a) Collection places of wild and cultivated *Hevea* accessions. The circles in different diameter represented the different number of collected accessions. The star indicated the location from where the Wickham germplasms were collected. Red, green, and blue cycles represented the wild germplasms from Acre, Rondonia and, Mato Grosso, respectively. The purple cycles represented the cultivars. (b) Phylogenetic tree of *Hevea* accessions with *H. spruceana* as an outgroup. The different colors of the circular perimeter indicated the group name of each clade. (c) Structure analysis with $K = 2, 3$, and 4 . (d) PCA plot of the first three components (PC1, PC2 and PC3). (e) Population divergence and genetic diversity across the four clades. The values in the nodes represented the genetic diversity (π) of the clades (purple, blue, red, and green nodes represented Cul, WMG, WAC and WRO clades, respectively), and the values between the nodes indicated population divergence (F_{st}). (f) Estimation of LD decay-distance of the four clades. (g) Latex production of the four clades.

Supplementary Fig. 4 Population structure analysis with $k = 2$ to 10. The accessions in Cul diverged obviously when $K > 4$.

Comment 5: “FST and pi can be computed only for panmictic populations without admixed samples: as FST and pi rely on the estimate of neutral site frequency spectrum; the occurrence of admixed individuals skew the SFS and then the selection inferences. **The authors must therefore remove the admixed individuals and define the population of “pure” individuals = group of individuals assigned with a membership coefficient > X% to a given cluster.**”

Response 5: Many thanks! In fact, we have removed the so-called admixed individuals which were located far from the major clades in the phylogenetic tree or mixed into the clade of different collection source before we performed FST and pi analyses. We are sorry for the vague description of “clade WAC” and “WAC”, “clade WRO” and “WRO”, “clade WMG” and “WMG”, and “clade Cul” and “Cul” between the main text and Fig. 2 that causes the misunderstanding. The WAC, WRO, WMG, and Cul represent the accessions without removing the admixed individuals. They are renamed clade WAC (5 admixed individuals were removed from 46 wild accessions in WAC), clade WRO (17 admixed individuals were removed from 81 wild accessions in WRO), clade WMG (22 admixed individuals were removed from 81 wild accessions in WMG), and clade Cul (11 admixed individuals were removed from the 127 cultivars in WMG) after the admixed individuals being removed. We have revised accordingly in the revised manuscript.

Comment 6: “In addition, pi and FST have their own limits, and the authors must use, at least, an additional statistic. For instance, the composite likelihood ratio (CLR) statistic and/or the mu statistics (<https://github.com/alachins/raisd>).”

Response 6: Many thanks! The XP-CLR statistic was used in addition to pi and FST in the revised version. A total of 22.56 Mb of genomic sequences covering 1,804 genes were detected using the XP-CLR method (Supplementary Table 15). In the list, there are eight JA signaling-related genes (*HbAOC2*, *HbLOX17*, *HbOPR8*, *HbOPR10*, *HbMYC24*, *HbHAT3*, *HbMAPKKK35*, and *HbMAPKK2*), five rubber biosynthesis-related genes (*HbACAT4*, *HbHbFPS1*, *HbGPS4*, *HbSRPP4*, and *HbREF1*), eleven ethylene signaling-related genes (*HbEIN4*, *HbERF2*, *HbERF19*, *HbERF20*, *HbERF25*, *HbERF26*, *HbERF48*, *HbERF61*, *HbERF100*, *HbERF142*, and

HbERF145) and eight ROS scavenging-related genes (*HbPOD7*, *HbPOD50*, *HbPOD91*, *HbGST20*, *HbGST21*, *HbGST50*, *HbGSH-Px3*, and *HbAPX2*). We have added these data in the main text (Fig. 3) and Supplementary Table 15 in the revised version.

Fig. 3 Genome-wide distribution of selective sweeps on the CATAS8-79 genome and selective features. (a) Selective sweeps detection by π , XP-CLR, and *Fst* methods. (b-d) Changes in nucleotide diversity (π) and enrichment of haplotypes of *HbOPR2* (b), *HbFPS1* (c) and *HbACS4* (d) across three generations and the correlations of haplotypes with corresponding phenotypes (NLR and latex production). Pri, primary clones; Sec, secondary clones; Ter, ternary clones; Hap, haplotype. n, the number of accessions.

Comment 7: “GWAS should also be run without the admixed individuals.”

Response 7: Thanks! In our study, the population structure had been corrected by kinship matrix generated with the FaST-LMM program, which could be considered to eliminate the influences on GWAS analysis of the so-called admixed individuals to a large extent. We have provided more information in Methods in the revised manuscript.

Comment 8: “The authors talk about “domestication” of rubber trees that started 150 years ago. Can we talk about domestication in the case of the rubber tree, or only of breeding? A discussion on that part may be appreciated.”

Response 8: Thank you for your interesting suggestion. Indeed, there is a viewpoint that wild plants are domesticated to be crops by artificial selection and the improvement of crops is performed by breeding, especially in the case of annual crops (Huang X, Huang S, Han B, Li J. The integrated genomics of crop domestication and breeding. *Cell*, 2022, 185: 2828-2839). In the case of rubber tree, however, the domestication characteristics are obvious based on the following reasons: (1) Plant domestication is a broad concept. It is a dynamic and ongoing process of adapting a wild species genetically to human use by selecting specific traits (Meyer RS., DuVal AE., Jensen HR. Patterns and processes in crop domestication: an historical review and quantitative analysis of 203 global food crops. *New Phytol.*, 2012, 196: 29-48; Gaut BS., Díez CM., Morrell PL. Genomics and the contrasting dynamics of annual and perennial domestication. *Trends Genet.*, 2015, 31: 709-719); (2) The cultivated rubber tree undergoes a process that generally fits in with the model of domestication of perennial crops (“wild forests---selection of outstanding genotypes---clonal propagation---cultivation”) (Gaut BS., Díez CM., Morrell PL. Genomics and the contrasting dynamics of annual and perennial domestication. *Trends Genet.*, 2015, 31: 709-719). The Wickham collections represent the wild germplasm of rubber tree. People select the individuals with good productivity from the natural population derived either directly from the Wickham collections or their progeny by open pollination, propagate by bud grafting, cultivate, and further disperse to other regions,

leading to local adaption. The representatives are the selection and application of rubber tree cultivars PR107, PB86 and GT1; (3) The artificial selection within 150 years has changed the *Hevea* cultivars morphologically (more laticifer rings and more latex production) and genetically (detected by PCA and population structure) although only less than four genetic recombinations occurred. We have provided the related information about domestication in Introduction section in the revised version.

Comment 9: “Line 38: precise genome size”.

Response 9: The precise genome size is 1,583 megabases. We have added it in the Abstract section of the revised version.

Comment 10: “-Line 38: “upgraded“ here, but in the main text, it sounds like it is a new genome? Please rephrase, correct”.

Response 10: Thank you. It is a new genome. We have revised.

Comment 11: “Line 44: PSK, please define for a non-specialist.”

Response 11: PSK is an abbreviation of phytosulfokine. It has been defined in the revised version.

Comment 12: “Line 46: *Taraxacum kok-saghyz*, unclear for non-specialist, explain.”

Response 12: Thanks, we inserted “Russian dandelion”, the English name of the species, before the Latin names “*Taraxacum kok-saghyz*” in the revised version.

Comment 13: “Line 50: One paragraph which introduces more broadly the questions (see major comments) would be a plus.”

Response 13: Thank you for your valuable suggestion. It has been revised accordingly. Please refer to **Response 1**.

Comment 14: “Line 67: 150 years, 5 generations then, even less when you take in account clonal propagation? Add this information? ”

Response 14: Thank you for your good suggestion. It has been revised accordingly. The sentence has been rephased as the followings: “The difficulty to dissect the component

traits of yield as well as the long asexual period, however, make the traditional breeding process of rubber tree time-consuming (> 20-25 years)¹⁴, resulting in less than four times of manual guidance sexual recombinations in *Hevea* cultivars within 150 years to date”.

Comment 15: “Line 69: What do you mean by “genuine clonally propagated crop”? please rephrase.”

Response 15: We have rephrased this sentence as the followings: “Nevertheless, the individual with favorite traits can easily be retained and propagated by bud grafting”.

Comment 16: “Lines 97-100: In the abstract you say “upgraded”, but here it sounds like it is a new genome? Please rephrase, correct either here or in the abstract.”

Response 16: The "upgraded" has been rephrased as "a high-quality" in the abstract.

Comment17: “Line 102: correct 1,583Mb, by 1.53 Gb (to make the parallel with line 99).”

Response 17: We have corrected “1,583Mb” by 1.58 Gb as well as the “1,554Mb” by 1.55 Gb in the revised version.

Comment 18: “Line 112: “repeats”, precise, is it only transposable elements (TEs), or also SSR or? And what is the proportion of TEs in the genome? This information should be added in Table 1.”

Response 18: The "repeat" covers TEs, low complexity repeat, tandem repeat, and unclassified repeat. The proportion of TEs in the genome is 61.59% (Supplementary Table 6). The detailed information has been provided in revised MS.

Supplementary Table 6 Annotation of repeats in CATAS8-79 assembly

	Length (bp)	Percentage of genome
Class I:Retrotransposon	946,288,831	60.88%
LTR-Retrotransposon	929,105,147	59.77%
LTR/Copia	229,070,420	14.74%
LTR/Gypsy	686,195,227	44.14%
LTR-other	13,839,500	0.89%
Non-LTR Retrotransposon	17,183,684	1.11%
SINE	78,335	0.01%

LINE	17,105,349	1.10%
Class II:DNA Transposon	11,094,572	0.71%
EnSpm/CACTA	2,804,304	0.18%
Harbinger	374,682	0.02%
Helitron	2,083,689	0.13%
MuDR	2,412,732	0.16%
Tcl/Mariner	285,790	0.02%
Hat	2,609,166	0.17%
DNA-other	524,209	0.03%
Low Complexity	5,565,222	0.36%
Tandem repeat	42,508,572	2.73%
Unclassified	174,661,477	11.24%
Total content	1,180,118,674	75.92%

Comment 19: “Line 118: unclear what does “CATAS” mean without any information before. Please define.”

Response 19: The CATAS means Chinese Academy of Tropical Agricultural Sciences. It has been defined as “CATAS8-79 (a rubber tree clone selected by Chinese Academy of Tropical Agricultural Sciences)” in the revised version.

Comment 20: “Line 152: the authors should present the STRUCTURE outputs in the main text and remove the admixed individuals for selection analyses (see major comments).”

Response 20: We have revised accordingly. Please refer to **Response 4** and **Response 5**.

Comment 21: “Line 160: “decrease of genetic diversity ... food crop”. That depends, see Gaut et al 2015 and Cornille et al 2014 (reference above), actually you do not expect that in trees. Please rephrase. ”

Response 21: Thanks. We have rephrased as the followings in the revised version: “The nucleotide diversity (π) of rubber tree cultivars (5.83×10^{-3}) was not significantly lower in comparison to that of the wild germplasms ($5.85 \times 10^{-3} \sim 7.12 \times 10^{-3}$, Fig. 2e), which was different from the remarkable decrease in genetic diversity in seed-propagated annual crops²⁷. The little decrease in genetic diversity of rubber tree cultivars should be mainly ascribed to the bud grafting propagation, which could contribute to retain the genetic diversity in many perennial crops².”

Additionally, the fewer occurrences of sexual recombinations and the large size and high heterozygosity of the rubber tree genome may also help to maintain genetic diversity.”.

Comment 22: “line 165: “cause by fewer occurrence of sexual recombination”. It sounds counterintuitive, you should expect that few recombinations lead to lower diversity... Please rephrase.”

Response 22: Thanks. We mean that few recombinations can prevent the decrease in the diversity. We have revised. Please refer to **Response 21**.

Comment 23: “line 196: and why not using as well sweep regions obtained with pi? ”

Response 23: Thank you for your kindly suggestions. We had also analyzed the sweep regions obtained by pi, but found no sweep regions overlapped with the GWAS strong signal.

Comment 24: “line 222: which genotypes did you use? ”

Response 24: The genotype we used is rubber tree clone CATAS7-33-97. We have revised in the revised version.

Comment 25: “Figures : all figures should be self-explanatory, i.e. with species names, number of samples used, and if needed, method used.

-Figure 1: add species names, number of accessions (wild and crop? How many?) and any relevant data used to make the figure (assembly, genome size, n50...). ”

-Figure 1: (VI) gene expression level for different organs, all organs pooled? Precise.

Response 25: Thank you for your kindly suggestions. We have accordingly revised as the followings: “Features of rubber tree clone CATAS8-79 genome with an assembly size of 1.58 Gb and a contig N50 of 11.21 Mb. (I) 18 chromosomes; (II) density transposon elements in a 100-kb sliding window; (III) density of copia retrotransposons in a 100-kb sliding window; (IV) density of gypsy retrotransposons in a 100-kb sliding window; (V) gene density in a 100-kb sliding window; (VI) gene expression values represented by the highest expression level in any of seven tissues based on $\log_2(\text{FPKM}+1)$ of RNA-seq data; (VII) distribution of guanine-cytosine content in a 100-kb sliding window, average GC content reached to 34.62%; (VIII) Intragenomic

syntenic regions covering at least 20 paralogues.” .

Comment 26: “Figure 1: (IV) Why did the authors present copia of only retrotransposon and gypsy? Why not showing all the TEs? ”

Response 26: Considering that the other TEs take very low percent of the repeats with appearance of nearly blank track, only the abundant copia and gypsy were presented.

Comment 27: “-Figure 4 explains a lot of information. Can’t it be split? In addition the number of samples used is unclear. For (e) 208 cultivated and wild germplasm? and (f) 127 cultivars and 208 wild germplasm. Please make it clearer. ”

Response 27: Thanks! We think it is better to be integrated in the present form. For “(e)208 cultivated and wild germplasm”, we have rephrased as “(e) ... 93 cultivated and 115 wild germplasms.” in the revised version.

Point-by-point responses to Reviewer 2' comments

Comment 1: “First, I was bothered by the fact that the generated data (raw reads, resequenced accesions, RNA-Seq data) is not available. Also, genome assembly and annotation could not be downloaded. Therefore, I could not verify the quality of the scientific work. Authors must share the data with the editor and reviewers and ensure that the data will be available when the article is published.”

Response 1: Thanks. The re-sequence and the RNA-Seq data can be downloaded by the following URL: <https://ngdc.cncb.ac.cn/gsa/s/PKGRVkHzH>, and <https://ngdc.cncb.ac.cn/gsa/s/WC128irX>. The genome assembly and annotation were uploaded to the open dissemination research data repository Zenodo with a website: <https://doi.org/10.5281/zenodo.7123623>.

Comment 2: “More generally, the article contains frequent shortcuts and lacks explanations which complicate understanding and raise doubts on the methods used by the authors.”

Response 2: Thanks! We have examined the shortcuts in detail and given brief explanations for the shortcuts in the revised version.

Comment 3: “Several years ago, two important loci related to latex production (LOL, Lot Of Latex) were identified (<https://doi.org/10.1111/nph.15253>). The authors also established a genetic link between jasmonic acid (JA) signaling and latex production in laticifers. I wondered if these genomic regions and associated genes had homologs in rubber-producing plants.”

Response 3: Many thanks! We carefully studied the paper you mentioned. The authors detected two mutants (*lol2* and *lol5*) with enhanced latex production from *Euphorbia lathyris* population mutagenized by gamma ray. They further confirmed that *lol2* and *lol5* manifested as Mendelian recessive genes and were not allelic. The enhanced latex production was caused by activating latex metabolism in *lol2* while by increased laticifer index (laticifer growth or differentiation) in *lol5*. A comprehensive identification of the mutant *lol2* had performed. The results showed a 100-fold increase in the basal accumulation of JA-Ile that was associated with the early steps of

the JA biosynthesis based on the significantly upregulated of LOX2 and AOS genes. Unfortunately, neither the genomic regions nor the genes that conferred the *lol2* and *lol5* mutants was identified in the study.

Comment 4: “The authors annotated several PSK genes in the Hv genome, one would expect to have a comparative analysis that could trace the evolutionary history of this important gene family.”

Response 4: As suggested, we analyzed the presence of the PSK gene family in 20 species (Supplementary Table 22). It is noted that PSK genes are not found either in *Sphagnum palustre* belonging to bryophyte or in *Selaginella tamariscina* belonging to pteridophyte but present in all the species belonging to spermatophyte. We further construct a phylogenetic tree using RAxML with 1,000 bootstrap replicates. Phylogenetic analysis showed that the PSK members could be divided into two clades. Clade I contain the members from spermatophyte while clade II is specific for angiosperm. The PSK members are more conservative in gymnosperm than in angiosperm while diverge more seriously in monocotyledon than in dicotyledon (Supplementary Fig. 7). We have added these data in the revised version.

Supplementary Table 22 PSK genes in plants

Phylum	Class	Order	Family	Genus	Species	Number of PSK
Bryophytina	Sphagnopsida	Sphagnales	Sphagnaceae	Sphagnum	Sphagnum palustre	0
Pteridophyta	Lycopodiinae	Selaginellales	Selaginellaceae	Selaginella	Selaginella tamariscina	0
Gymnospermae	Coniferopsida	Coniferales	Pinaceae	Thuja	Thuja plicata	4
		Amborellales	Amborellaceae	Amborella	Amborella trichopoda	2
		Ranunculales	Nymphaeaceae	Nymphaea	Nymphaea colorata	5
Angiospermae	Dicotyledoneae	Urticales	Eucommiaceae	Eucommia	Eucommia ulmoides	3
		Urticales	Moraceae	Morus	Morus notabilis	5
		Rosales	Rosaceae	Malus	Malus domestica	10

	Campanulales	Compositae	Taraxacum	Taraxacum kok-saghyz	4
	Tubiflorae	Solanaceae	Solanum	Solanum tuberosum	8
	Tubiflorae	Solanaceae	Solanum	Solanum lycopersicum	7
	Rhamnales	Vitaceae	Vitis	Vitis vinifera	5
	Salicales	Salicaceae	Populus	Populus trichocarpa	7
	Geraniales	Euphorbiaceae	Hevea	Hevea brasiliensis	6
	Geraniales	Euphorbiaceae	Manihot	Manihot esculenta	7
	Geraniales	Euphorbiaceae	Jatropha	Jatropha curcas	5
	Geraniales	Euphorbiaceae	Ricinus	Ricinus communis	4
	Papaverales	Cruciferae	Arabidopsis	Arabidopsis thaliana	6
Monocotyledoneae	Graminales	Gramineae	Oryza	Oryza sativa	7
			Zea	Zea mays	7

Supplementary Fig. 7 Phylogenetic tree of PSK proteins from 18 species. Black triangle represents *Thuja plicata* (Gymnospermae). Colored circle represents laticifer-contained plants. Red font shows *HbPSK5*.

Comment 5: “The abstract contains undefined abbreviations (MYC, JA, PSK), repetitions and unclear statements (by promoting laticifer formation in *Taraxacum kok-saghyz*). ”

Response 5: We have revised accordingly in the revised version.

Comment 6: “L100. The genome size was estimated by the kmer distribution but Supplementary Table 1 does not contain any size estimate. According to the methods section, the authors use Genomescope, but they must provide distribution plots and results.”

Response 6: Thank you. We have used Genomescope to estimate the genome size. It seems to be unsuitable for estimating the size of the genome with high heterozygosity such as the rubber tree genome (please refer to the following figure1). Instead, Jellyfish software is used to estimate the genome size mainly based on the kmer distribution, and the result is extremely consistent with the size of our genome assembly (please refer to the following supplementary Table 1 in the revised version).

Figure1 The histogram of k-mer 17.

Supplementary Table1 Estimate of CATAS8-79 genome size

Total sequence read number	The average length of sequence read (bp)	K-mer length	Total number of low frequency K-mer	The overall depth estimated from K-mer distribution	Genome Size
570,529,388	150	17	1,282,885,189	50	1,503,361,055
570,529,388	150	19	1,945,729,055	47	1,503,361,056
570,529,388	150	21	2,265,914,525	45	1,597,842,354
570,529,388	150	23	2,487,322,738	44	1,603,191,794
570,529,388	150	25	2,670,043,644	43	1,609,689,705
570,529,388	150	27	2,820,662,545	42	1,617,261,466

Comment 7: “L100. Please describe the type of PACBIO data that was generated.”

Response 7: The type of PACBIO data is PacBio CLR (continuous long-read).

Comment 8: “L104. Supplementary Figure 1 should be a table.”

Response 8: Thanks. “Supplementary Figure 1” has been split into three tables and removed to Supplementary Tables (Supplementary Table 3, Supplementary Table 5, Supplementary Table 6) in the revised version.

Comment 9: “I wonder what is the heterozygosity rate of the rubber tree genome. I suspect that this rate is higher than 1%.”

Response 9: The heterozygosity rate of the rubber tree genome is 1.26% estimated by Jellyfish software. We have added it in the revised version.

Comment 10: “The authors use the Pilon polisher which has been described to perform poorly in the case of heterozygous genomes and can lead to fragmented genes (<https://doi.org/10.1093/nargab/lqab034>).”

Response 10: Thank you very much! It is an ongoing process to improve the quality of genome assembly. We will provide a higher quality of haplotype assembly of rubber tree clone CATAS8-79 by your suggested method (Aury and Istace, 2021) in the future.

Comment 11: “Also, the BUSCO score is higher for genome assembly than for gene models, and the difference is due to a higher proportion of fragmented genes, which is

linked to the point raised earlier. Moreover, the proportion of duplicated genes is high, does it mean that the final assembly still contains allelic duplications? The authors should provide more information to the reader.”

Response 11: Many thanks! In this study, the proportion of "Complete and duplicated BUSCO genes" is 10.50%, which is consistent with and even almost equal to the ratio of duplicated BUSCO genes (9.9%) in the released GT1 rubber tree genome (please refer to the following Table1) (Liu et al. The chromosome-based rubber tree genome provides new insights into spurge genome evolution and rubber biosynthesis. *Mol. Plant*, 2020, 13, 336-350). Most of the high-confidence genes were duplicated genes. We also found that most of the Ks values were included in the two peaks (please refer to the following figure2), which further supported that most duplicated genes originated from the WGDs, and these results were extremely consistent with the reported recent specific WGD in rubber tree (Tang et al. The rubber tree genome reveals new insights into rubber production and species adaptation. *Nature Plants*, 2016, 2:16073; Liu et al. The chromosome-based rubber tree genome provides new insights into spurge genome evolution and rubber biosynthesis. *Mol. Plant*, 2020, 13: 336-350). Therefore, all these results showed that the slightly higher ratio of duplicated genes in rubber tree might be due to the WGD events. We have provided the information in the revised version.

Table1 BUSCO analysis of two genomes

	CATAS8-79 (In this study)	GT1 (Liu et al., 2020)
Complete BUSCOs (C)	95.8%	93.5%
Complete and single-copy BUSCOs (S)	85.3%	83.6%
Complete and duplicated BUSCOs (D)	10.5%	9.9%
Fragmented BUSCOs (F)	2.1%	2.7%
Missing BUSCOs (M)	2.1%	3.8%

Figure2 Ks value of CATAS8-79

Comment 12: “L116. High-confidence gene models, how confidence was assessed ?”

Response 12: We firstly establish an "Allset" based on *ab initio* predictions. High-confidence gene models "termed as Coreset" are further assessed by removing partial gene models and gene models with an internal stop codon, a hit to a transposable element, or an FPKM of 0 across the RNA-Seq/Iso-seq data used for the annotation. Genome annotation of Allset and Coreset are provided (the following table2). We have modified it in the "Methods" section.

Table2 Genome annotation of allset and coreset

Feature	Allset	Coreset
Gene count	102,356	38,595
Gene max (bp)	273,469	75,969
Gene min (bp)	102	102
Gene Median/AVG (bp)	4,381/5,599.87	3,680/5,282.54
mRNA count	102,428	82,641
mRNA max (bp)	21,964	21,964
mRNA min (bp)	102	102

mRNA Median/AVG (bp)	1,047/1,405.43	1,396/1,680.61
mRNA count Median/AVG	1/1.00	1/1.00
CDS Median/AVG (bp)	981/1323.25	1,023/1,275.52
Protein Median/AVG (aa)	327/441.08	341/425.17
exon Median/AVG (bp)	110/187.89	130/217.09
exon count Median/AVG	6/7.47	6/7.74
UTR3 Median/AVG (bp)	269/406.23	137/211.01
UTR5 Median/AVG (bp)	73/179.26	95/154.07
Intron Median/AVG (bp)	281/610.86	201/552.46
Intron count Median/AVG	5/6.47	5/6.74

Comment 13: “L119-120, the link between the number of syntenic blocks and higher contiguity is unclear.”

Response 13: Thanks. To avoid misunderstanding, we have deleted this sentence in revised version.

Comment14: - L127. I assume Hb stands for *Hevea brasiliensis*, but please add that information.

Response 14: Yes, the information had added as “A total of 26 HbMYCs were identified from *Hevea brasiliensis*” in the revised version.

Comment 15: “ L132. Clade A, please add information on existing clades.”

Response 15: We have added the information (Supplementary Fig. 3) in the revised version.

Supplementary Fig. 3 Identification of *HbMYC* gene family in CATAS8-79 genome. (a) Motif analysis of *HbMYCs* using the MEME program. The red motif represented bHLH domain while the green motif represented MYC domain. (b) Location and interchromosomal relationships among *HbMYCs* in CATAS8-79 genome. Coloured lines indicated syntenic blocks. (c) Phylogenetic tree of MYC proteins. I-VII shows seven clades. At: *Arabidopsis thaliana*; Os: *Oryza sativa*; Pt: *Populus trichocarpa*; Tk: *Taraxacum kok-saghyz*; Me: *Manihot esculenta*; Hb: *Hevea brasiliensis*. (d) Synteny analysis of four species showing the expansion of MYC members. (e) Expression patterns of 26 *HbMYCs* in the latex from virgin trees (VL) and tapped trees (TL), and in the cambial region upon COR treatment for 0.5 h (COR/0.5 h) and 1 h (COR/1 h). The bar indicated the relative expression levels by color.

Comment 16: “L145. Supplementary Table 8 contains a column titled “coverage”. I assume the value represents the fraction of the reference genome that is covered. This coverage value is often close to 70-80%. How do the authors explain these low values ? ”

Response 16: Thanks. To ensure the SNP accuracy for population genetic analysis, we had filtered the mapping reads with low-mapping quality (MAQ<30), which resulted in the low coverage to the reference genome. The coverage values were high, ranging from 90.26% to 94.67% if the best match for every read was retained without MAQ<30 filtering (the below Table 3).

Table 3 Statistical analysis of coverage ratio included or excluded low-mapping quality reads

Sample	Low-mapping quality reads	Low-mapping quality reads
	included	excluded
Cul-1	94.47%	82.26%
Cul-11	94.23%	81.71%
Cul-18	93.96%	81.09%
Cul-81	94.63%	82.60%
WAC-12	94.07%	81.28%
WAC-16	91.09%	69.63%
WAC-23	90.40%	68.25%
WAC-5	90.26%	68.43%
WMG-15	93.57%	79.61%
WMG-23	93.86%	80.09%
WMG-4	94.67%	82.61%
WMG-8	92.65%	78.11%
WRO-15	91.24%	70.55%
WRO-2	90.85%	69.80%
WRO-24	90.75%	69.94%
WRO-35	92.21%	71.87%

Comment 17: “L189-190. “The expression patterns of the selected genes were associated with yield productivity”. Authors need to go further and perform correlation analysis if they want to associate expression patterns with traits.”

Response 17: Thank you for your valuable suggestions. We have rephased and given a detailed description in the revised version as the followings: “These selected genes might confer the higher NLR and rubber biosynthesis efficiency and longer duration of latex flow in cultivars in comparison with the wild relatives (Supplementary Fig. 5). The expression of *HbOPR2*, encoding a homolog of *AtOPR3* that is a key enzyme for endogenous JA biosynthesis²⁹, was rapidly up-regulated in the cambia region in response to exogenous coronatine, which may be associated with the biosynthesis of endogenous JA and the secondary laticifer differentiation³⁰ (Supplementary Fig. 5a-b). The expression levels of *HbNIN3* and *HbFPS1* were remarkably up-regulated (Supplementary Fig. 5c), which may be associated with the enhanced latex regeneration in regularly tapped tree¹⁷. The *HbACS4* was mainly up-regulated in the

inner bark of tapped trees (Supplementary Fig. 5d), which appears to confer the endogenous ethylene production upon tapping, resulting in a prolongation of duration of latex flow. A higher transcriptional level of *HbSOD4* in the rubber tree clone CATAS8-79 (Supplementary Fig. 5e) may be associated with the increased duration of latex flow in comparison with rubber tree clone RP107³¹.”.

Comment 18: “L335. Please provide optical maps information, Supplementary Table S2 contains raw data metrics only.”

Response 18: We have provided sequenced optical maps information in the revised version (Supplementary Table 2) as the followings.

Supplementary Table 2 BioNano information for CATAS8-79 assembly

Feature of sequenced optical maps	
Basic data generated by BspQI enzyme digestion	
Quantity (Gbp)	206.85
Avg. N50 (Kbp) (>= 150 Kbb)	300.8
Avg. N50 (Kbp) (>= 20 Kbb)	264.7
Avg. Label Density (per 100 Kbp)	8.52
Avg. Map Rate (%)	53.70%
Optical map assembly	
Number Genome Maps	1109
Total Genome Map Length (Mbp)	2701.09
Mean Genome Map Length (Mbp)	2.436
Median Genome Map Length (Mbp)	1.532
Genome Map N50 (Mbp)	3.801
Total Reference Length (Mbp)	2088.074
Total Genome Map Length/Reference Length	1.294
Total number of aligned Genome Maps	1098 (0.99)
Total Aligned Length (Mbp)	2320.746
Total Aligned Length/Reference Length	1.111
Total Unique Aligned Length (Mbp)	1650.289
Total Unique Aligned Length/Reference Length	0.79

Comment 19: “I could not find any information on how the RPKMs in Supplementary Table S6 were calculated.”

Response 19: Thanks. Because the cDNA libraries were paired-end sequenced on the BGISEQ-500 platform, the RPKM should be FPKM (Fragments Per Kilobase of transcript per Million mapped reads). We have revised it in Methods section, figure

legends and Supplementary Table S9 in the revised version.

Comment 20: “I suggest adding information about the current rubber tree reference genome in Table 1 along with the BUSCO and Merqury scores for both assemblies.”

Response 20: Many thanks. Based on your suggestions, we have evaluated the quality of the current assembly using Merqury software. Results showed that the value of k-mer completeness was 82.1%, while the value of consensus quality value (QV) was 28.9 (the below Table 4). These scores were lower than that of *Thlaspi arvense* genome published recently (Nunn et al., Chromosome-level *Thlaspi arvense* genome provides new tools for translational research and for a newly domesticated cash cover crop of the cooler climates. Plant Biotechnol J. 2022 May; 20(5): 944–963). The difference may be likely caused by different sequencing technologies. In comparison to previous sequence technology Pacbio-CLR, the recent Pacbio-HIFI had higher fidelity and improved genome accuracy. We believed the new technology will promote the high-accuracy rubber tree haplotype assembly in the future.

Table 4 Merqury k-mer analysis of total assembly showing relative completeness of k-mers present in each read set from Illumina HiSeq.

k-mers (asm)	k-mers (reads)	% completeness	QV
475,861,177	579,572,561	82.1055	28.9199

Comment 21: “Figure 1: a track is missing”

Response 21: Thanks. We have revised in the revised version.

Comment 22: “- Figure 2c : “cycles” should be replaced with “nodes”, and “between the cycles” with “edges”. ”

Response 22: Many thanks. We have revised accordingly.

Comment 23: “- Figure 2d. How was the difference in latex production measured ? ”.

Response 23: We have rephrased the sentence as “Latex production of the four clades” .

Comment 24: “- Figure 4a. The Manhattan plot shows a significant signal on chromosome 15, as noted by the authors. However, they did not mention the signal on chromosome 13.”

Response 24: It is reasonable to focus on the most significant signal on chromosome 15. We did not mention the signal on chromosome 13 because the signal was not obvious.

Comment 25: “- Figure 4c. The scale is required. Also the figure legend mentions FPKM values, but Table S6 contains RPKM.”

Response 25: The scale of Figure 4c has added in the revised version. The “RPKM” in Table S6 was indeed a misspelling, we have corrected it to “FPKM” in Supplementary Table S9 in the revised version.

Point-by-point responses for the Reviewer 3’ comments

Comment 1: “Even though PSK has been implicated in vascular cell fate control before, its proposed function in promoting laticifer cell fate is certainly novel and very interesting. However, the data presented here do at present insufficiently support a role for PSK in rubber production.”

Response 1: Thank you for your affirmative. Available data showed that the enhanced latex production could be caused by increased number of laticifer cells (Castelblanque et al. *LOL2* and *LOL5* loci control latex production by laticifer cells in *Euphorbia lathyris*. *New Phytol.*, 2018, 219: 1467-1479.). We have rephased the sentence as the followings: “Overexpression of *HbMYC26* and *HbPSK5* in TKS promoted laticifer formation, causing 14.29% ~ 49.38% and 15.48% ~ 76.77% increases in rubber contents, respectively.” in the Results of the revised version.

Comment 2: “The correlation of genotype and rubber production is high and convincing but it is insufficiently explained how PSK5 might affect rubber production. The associated SNP itself is 700 bp upstream of the transcriptional start site. It is not

clear whether the authors assume this also is the causative mutation. The manuscript is very vague on this point. Would associated indels be detected by their analyses? Were PSK5 alleles from different accession sequenced? If the SNP is indeed supposed to be causative, the plausible scenario is that PSK5 expression differs between genotypes. However, this is not experimentally verified by, e.g. by QPCR analyses in a few accessions. A mutation like this is a priori rather unlikely to have a large scale effect and PSK5 expression in the elite cultivar CATAS8-79 is rather low compared to the other PSKs (possibly due to tissue specificity), therefore more efforts to strengthen the causal link are warranted. Related with the previous point, the transgenic TKS lines are overexpressors of the MYC transcription factor or the PSK target. Ideally one would compare the different alleles but obviously the upstream regulatory sequences from rubber tree can't be expected to yield insight in the heterologous system. However, while this limitation is acknowledged, it places even higher emphasis on the need for correlative PSK5/MYC expression analysis in accessions with different genotypes/rubber production.”

Response 2: Thank you for your valuable suggestions. We have performed experiments to address your concerns. We detected 30 SNPs and no Indel within 1500 bp gene region of *HbPSK5* (see below figure 3) in 12 accessions with phenotype of higher NLR and 11 accessions with phenotype of lower NLR by Sanger sequencing. Among the SNPs, only the SNP at -760 bp is highly associated with the phenotype. We also analyzed the expression patterns of *HbPSK5* in the germplasm with GG or AA genotype in the cambium region obtained by frozen section. It appeared a tendency that the expression levels were higher in GG genotype than in AA genotype germplasm (see below figure 4). However, we could hardly conclude because of its rather low expression level. The low expression level might be related to its cell-specific expression, because *HbPSK5* only expressed in few cells in the cluster related to laticifer differentiation by single-cell RNA sequencing on rubber tree cambium region (see below figure 5).

Figure 3 Relationship between SNP and NLR among 23 individuals. Red star shows the significant SNP.

Figure 4 Expression levels of *HbPSK5* in GG genotype (Reyan72059-1/2/3; Dafeng95-1/2/3; Reyan73397-1/2/3) and AA genotype (XJ000842-1/2/3; XJ006871-1/2/3; XJ003117-1/2/3) germplasm by q-PCR. The *HbUBC3* was used as a reference gene.

Figure 5 Expression patterns of *HbPSK5* on UMAP map. Number in the figure shows different clusters. Black arrow shows the cell expressed *HbPSK5*.

Comment 3: “Another concern with the transgenic lines is whether WT is the appropriate control. As far as I understand, TKS is transformed via leaf disc regeneration. It is not clear from the methods section whether these are regenerated or seed grown plants. Even if these are seed-grown, an empty vector-transformed transgenic line would be more appropriate than the WT. As regeneration manifests epigenetic changes and is mutagenic, a mock transformed line would rule out that increased laticifer cell production is unrelated to PSK/MYC expression. A line that only weakly expresses the respective transgene might also help here if it strengthens the correlation between laticifer cells and transgenes, as all the presented transgenic lines seem to show rather similar expression levels, which do not correlate with the phenotype.”

Response 3: Because the self-incompatible TKS line 1151 was used for transformation, the T₀ transgenic lines were propagated by tissue culture for further analysis. We have revised in the Methods section. Our experiment results showed that there was no obvious difference in the laticifer cell index between WT and empty

vector-transformed transgenic line (see below figure 6). The laticifer cell index was much less in *HbPSK5* or *HbMYC26* weakly expressed lines than that in *HbPSK5* or *HbMYC26* strongly expressed lines (please refer to the following Supplementary Fig. 9 and Supplementary Fig. 10 in the revised version).

Figure 6 The laticifer cell index and rubber content in the root of WT and empty vector-transformed transgenic lines. The statistical analysis was performed using Student's t-test (two-side) by GraphPad Prism v.8.4.2. All experiments were performed independently three times. Scale bars = 1 mm.

Supplementary Fig. 9 Functional identification of *HbMYC26* in TKS. (a) Root cross sections of WT and *HbMYC26*-overexpressed lines. (b) Expression levels of *HbMYC26* in WT and *HbMYC26*-overexpressed lines. (c) Laticifer cell index in the root of WT and *HbMYC26*-overexpressed TKS lines. The statistical analysis was performed using Duncan's test by SPSS Statistics 17.05 for multiple group comparisons. All experiments were performed independently three times. The laticifer cells (blue arrowheads) were colored in deep red. Scale bars = 1 mm. The capital letter represents $P < 0.01$ while lower case represents $P < 0.05$. The same letter indicated no significant difference among groups.

Supplementary Fig. 10 Functional identification of *HbPSK5* in TKS. (a) Root cross sections of WT and *HbPSK5*-overexpressed lines. (b) Expression levels of *HbPSK5* in WT and *HbMYC26*-overexpressed lines. (c) Laticifer cell index in the root of WT and *HbPSK5*-overexpressed TKS lines. The statistical analysis was performed using Duncan's test by SPSS Statistics 17.05 for multiple group comparisons. All experiments were performed independently three times. The laticifer cells (blue arrowheads) were coloured in deep red. Scale bars = 1 mm. The capital letter represents $P < 0.01$ while lower case represents $P < 0.05$. The same letter indicated no significant difference among groups.

Comment 4: "Fig4I should be quantified"

Response 4: As suggested, we have quantified the number of laticifer rings in Fig 4i in the revised version.

Comment 5: "100 μ M of PSK seems rather high as a treatment, do lower doses show an effect as well? It is difficult to control how much PSK arrives at the site of action (presumably the cambium), my concern here is whether laticifer cell production might be a general stress response, somewhat in line with the observed increase by application of JA or by wounding. Lower doses would alleviate this concern, but I

fully acknowledge that a requirement for a high PSK dose does not imply a problem with the conclusion perse.”

Response 5: Thank you for your consideration and kindly suggestions. The laticifer cell production is not a general stress response considering that all the effective factors in inducing laticifer formation are directly or indirectly associated with jasmonates (Hao B.Z., and Wu J.L. Laticifer differentiation in *Hevea brasiliensis*: induction by exogenous jasmonic acid and linolenic acid. *Annals of Botany*, 2000, 85: 37-43). Wounding did not induce laticifer cell differentiation when the production of endogenous jasmonates was inhibited (Tian W.-M., Yang S.-G., Shi M.-J., Zhang S.-X., Wu J.-L. Mechanical wounding-induced laticifer differentiation in rubber tree: An indicative role of dehydration, hydrogen peroxide, and jasmonates. *Journal of Plant Physiology*, 2015, 182: 95-103). Moreover, application of a high level of exogenous ABA and ethephon (a releaser of ethylene), the stress signals, has no effect on inducing the secondary laticifer differentiation (see below Table cited from the reference “Hao B.Z., and Wu J.L. Laticifer differentiation in *Hevea brasiliensis*: induction by exogenous jasmonic acid and linolenic acid. *Annals of Botany*, 2000, 85: 37-43.”).

We have tested the effect of synthetic PSK with different concentrations (10 μ M, 100 μ M, and 500 μ M). The effect of 100 μ M PSK is the best of the three concentrations on inducing secondary laticifer differentiation although either the 10 μ M or 500 μ M PSK is somewhat effective.

TABLE 2. *Secondary laticifer differentiation as influenced by JA, SA, ABA and ethephon*

Treatment	Secondary laticifer line number	Secondary phloem cell layer number
0.04% JA	0.8 ± 0.5	9.5 ± 0.9
2% SA	0.1 ± 0.2	9.8 ± 0.2
0.2% SA	0	8.3 ± 1.1
0.5% ABA	0	8.5 ± 1.0
0.05% ABA	0	10.0 ± 1.3
2% Ethephon	0	8.3 ± 2.0
0.2% Ethephon	0	8.4 ± 2.5
Lanolin	0	10.0 ± 1.3

This table is cited from Hao and Wu, *Annals of Botany*, 2000, 85: 37-43.

Comment 6: “In line 124/125 a citation seems to be missing.”

Response 6: Thanks. We have revised in the revised version.

Comment 7: “Finally, a possibly naïve question from a GWAS non-expert: Given the few rounds of recombination during domestication, is it not very surprising to see that many significant loci in the Manhattan plot, suggesting many low impact variants?”

Response 7: Thanks. For a complex trait, there are usually many low impact associated QTLs which could be detected by GWAS. On the other hand, GWAS is a linkage-based analysis. Thus for each detected GWAS peak, there may be many significant non-causal SNPs linked to the (unknown) causal SNP and the causal SNP is not necessarily the most significant one.

Point-by-point responses to the Reviewer 4' comments

Comment 1: “Though I am happy that this represents a major contribution to *Hevea* genetics and biology I have some concern about the contrasts that have been made between the wild and cultivated genepools. This is for two reasons - first of all because of the area of the original Wickham collections is not represented in the wild component of the comparison and second because the cultivated germplasm is, inevitably for *Hevea*, heavily biased by over representation of several genotypes in the pedigrees. e.g., RRIM600 and PR107. However, though I believe that this does not negate the conclusions the authors make I would suggest that they need to acknowledge this in their conclusions. On this basis the identification of HbPSK5 as a key domestication gene might have been somewhat suspect. However, the functional analysis though transformation of *Taraxacum* goes a considerable way towards overcoming any such objection and I am happy to accept their argument.”

Response 1: Thank you for your valuable suggestions. Of course, it is better to perform contrast between cultivars and the wild germplasm around the area of the original Wickham collections. As you know, it is difficult to obtain such wild germplasm. Fortunately, population structure analysis shows that of the three wild populations, the wild germplasms from Mato Grosso are the closest relatives of cultivars which is further strengthened by the shared hydrographic net (seeds could float downstream of the river) between the area of Mato Grosso where the wild germplasm were collected by IRRDB in 1981 and the downstream area of the original Wickham collections. So, it should be reasonable to perform contrast between cultivars and the wild germplasm collected from Mato Grosso. As for your second reason, we acknowledge that the *Hevea* cultivars are biased by over representation of several genotypes, such as RRIM600, PR107 and PB86. To our knowledge, the over representation of cultivars by several genotypes (i.e., backbone parental) results in a genetic bottle, which is common for crop breeding and more heavily in annual crops than perennial crops. Maybe owing to the highly heterozygous genome, so far few rounds of sexual recombinations, and bud grafting propagation, the nucleotide

diversity of *Hevea* cultivars did not obviously reduce in comparison to their wild relatives, leading to serious trait segregation of F1 population. So, artificial selection is essential for the increased percentage of GG genotype in cultivars. In the wild populations, the genotype AA took a percentage of 93.27% while genotype GG of 0.96% based on the SNP in the promoter region of *HbPSK5*. In contrast to the case of a wild population, the percentage reduced to 29.13% for the AA genotype but increased to 23.62% for the GG genotype in cultivated population. For such reasons, the selection of *HbPSK5* is unlikely to be the result of over representation, but likely caused by artificial selection during domestication.

Comment 2: “On a technical point, I find many of the diagrams are too small to read. While I realise that there are severe constraints in terms of journal formats, at least for supplementary figures I see no reason for such small diagrams.”

Response 2: We have modified the diagrams format in supplementary figures.

Comment 3: “While I acknowledge the deposition of the sequence data and I can see that accessions have been created I am not able to access them as they are set to be released on the 1st January next year. I cannot therefore identify what in fact will be made public. I would expect genomic and transcriptomic sequence data, assembly data and annotated genes but it is not possible to verify from the text or online that this is indeed the case. I would also hope that the intention is to make the data available in a more tractable form for many researchers by making available in a browser format and would welcome a statement to that effect in the manuscript.”

Response 3: Thanks. The re-sequence data could be downloaded by the following URL: <https://ngdc.cncb.ac.cn/gsa/s/PKGRVkHzH>. The RNA-Seq data could be downloaded by the following URL: <https://ngdc.cncb.ac.cn/gsa/s/WC128irX>. The genome assembly and annotation were uploaded to the open dissemination research data repository Zenodo with a website: <https://doi.org/10.5281/zenodo.7123623>.

Reviewers' Comments:

Reviewer #1:

Remarks to the Author:

I have read the manuscript in its first version, which has been greatly improved. The authors of this manuscript investigated the genomic footprints of domestication in the rubber tree (*Hevea brasiliensis*) by combining a genome assembly and 335 resequenced accessions of wild and cultivated rubber trees. The authors have carefully addressed most of the comments, but I still have a major comment.

Major comment

The authors have now presented the population structure in the main text, which is much clearer. However, a substructure was observed in the cultivated gene pool for $K > 4$. The authors explained that this was because they had highly related individuals. If this is the case, the authors should have removed highly genetically related and duplicate individuals, for example, individuals with pairwise KING-robust kinship estimates among individuals > 0.354 , before running structure analyses. The authors should check the population structure when removing related individuals to check if they do not change their interpretation, and four clusters were still observed.

Minor comments

Note: There is a switch of cluster color between the bar plots from $K=2$ to $K=10$ in Figure S4. Please correct it.

-line 38: convert genome size in Gb

-lines 43-45: "The transcriptional activation of HbPSK5 by myelocytomatosis (MYC) members linked PSK signaling to jasmonic acid (JA) in regulating JA response, the laticifer differentiation in rubber tree." Please revise the sentence; it is tedious to understand.

-line 133: add "genotypes" after GT1

-lines 163-164: see my comments above on population structure

-line 186: "that artificial selection has worked during the domestication of rubber tree within 150 years". Please revise the wording. There was artificial selection during domestication [...].

-line 198: Please revise wording, "In the selective sweeps." Rather "Selective sweep were observed in 11 JA signalling-related genes.

-line 293: add "genetic" in the population GENETIC analyses.

Reviewer #2:

Remarks to the Author:

I have read the revised manuscript "Genomic insight into domestication of rubber tree" and congratulate the authors as the manuscript has been significantly improved. They answered many of my requests, but I still have a concern regarding the comparison of their assembly with existing rubber tree genome assemblies. Indeed, the article describes a genomic resource and potential users must have information on the quality of this resource compared to existing ones. Authors should provide a more comprehensive Table 1 with metrics (at least general statistics, BUSCO scores, Mercury completeness and QV) calculated on available genome assemblies.

Minor points:

- L448 : softwires -> software.

- Add Table2 (from reviewer's response) in the Supplementary Tables.

Reviewer #3:

Remarks to the Author:

The authors have addressed all my points and i agree with them that this version of the manuscript is indeed strongly improved. Several clarifications now also make the manuscript easier to follow. To reiterate, i am in no good position to judge the genomic aspects of the paper, but concerning the functional data, i have no further issues. If length restrictions permit, i would still suggest to help th reader with the dense content by adding a few sentences to connect the different aspects of the paper and elaborate on the functional experiments in the last part (essentially Fig. 4).

Responses to the Reviewers' comments on the MS (NCOMMS-22-20274A)

We here gratefully thank all the reviewers for the critical comments. The manuscript is further improved. All the revised are marked in red color. The followings are the point-to-point responses to the reviewers' comments on the MS (NCOMMS-22-20274A).

Responses to the Reviewer1's comments

Major comment: "The authors have now presented the population structure in the main text, which is much clearer. However, a substructure was observed in the cultivated gene pool for $K > 4$. The authors explained that this was because they had highly related individuals. If this is the case, the authors should have removed highly genetically related and duplicate individuals, for example, individuals with pairwise KING-robust kinship estimates among individuals > 0.354 , before running structure analyses. The authors should check the population structure when removing related individuals to check if they do not change their interpretation, and four clusters were still observed."

Response to the major comment: Thank you for your valuable suggestions. The population structure analyses were performed after removing the highly genetically related individuals. The four clusters were still observed (please refer to the figure 1).

Figure 1 Population structure analyses. (a) Population structure including the highly related individuals. (b) The pedigree of sub-population in Cul cluster. (c) Population structure excluding the highly related individuals.

Minor comment 1: “Note: There is a switch of cluster color between the bar plots from K=2 to K=10 in Figure S4. Please correct it.”

Response 1: Thank you for your kindly suggestions. We have adjusted the cluster color. As the differential color could hardly be displayed when $K > 7$, the population structures contained those with $k = 2$ to 7 in revised Figure S4.

Supplementary Fig. 4 Population structure analysis with $k=2$ to 7. The accessions in Cul diverged obviously when $K>4$.

Minor comment 2: “-line 38: convert genome size in Gb”

Response 2: We have corrected it.

Minor comment 3: “-lines 43-45: “The transcriptional activation of *HbPSK5* by myelocytomatosis (MYC) members linked PSK signaling to jasmonic acid (JA) in regulating JA response, the laticifer differentiation in rubber tree.” Please revise the sentence; it is tedious to understand.”

Response 3: Thanks for your suggestions. We have revised the sentence as “The transcriptional activation of *HbPSK5* by myelocytomatosis (MYC) members linked PSK signaling to jasmonic acid (JA) in regulating the laticifer differentiation in rubber tree.”

Minor comment 4: “-line 133: add “genotypes” after GT1.

Response 4: We have added it in the revised MS.

Minor comment 5: “-lines 163-164: see my comments above on population structure”

Response 5: We have reanalyzed the population structure. Please see **Response to the major comment**.

Minor comment 6: “-line 186: “that artificial selection has worked during the domestication of rubber tree within 150 years”. Please revise the wording. There was artificial selection during domestication [...]”.

Response 6: Thanks. We have corrected in the revised version.

Minor comment 7: “-line 198: Please revise wording, “In the selective sweeps.” Rather “Selective sweep were observed in 11 JA signalling-related genes.

Response 7: We have corrected in the revised version.

Comment 8: “-line 293: add “genetic” in the population GENETIC analyses.”

Response 8: We have corrected in the revised version.

Responses to Reviewer 2’ comments

Major comment: “I have read the revised manuscript “Genomic insight into domestication of rubber tree” and congratulate the authors as the manuscript has been significantly improved. They answered many of my requests, but I still have a concern regarding the comparison of their assembly with existing rubber tree genome assemblies. Indeed, the article describes a genomic resource and potential users must have information on the quality of this resource compared to existing ones. Authors should provide a more comprehensive Table 1 with metrics (at least general statistics, BUSCO scores, Mercury completeness and QV) calculated on available genome assemblies.”

Response to the major comment: Many thanks for your suggestions. The BUSCO scores, mercury completeness and consensus quality value has been calculated and the comparison of these metrics has performed between rubber tree clones CATAS8-79 and GT1, both of which were at chromosome-level. The three metrics were added in Table 1. The information on the comparison of BUSCOs between CATAS8-79 and GT1 assemblies was provided in Supplementary Table 4 and the comparison of mercury completeness and consensus quality value between CATAS8-79 and GT1 assemblies was provided in Supplementary Table 5. We have also modified information related to

the metrics in Results and Methods Sections in the revised version.

Minor comment 1: “- L448 : softwires -> software.”

Response 1: We have corrected it.

Minor comment 2: “- Add Table2 (from reviewer’s response) in the Supplementary Tables.”

Response 2: We have added it in Supplementary Tables as “Supplementary Table 8”.

Responses to Reviewer 3’ comments

Comments: “The authors have addressed all my points and i agree with them that this version of the manuscript is indeed strongly improved. Several clarifications now also make the manuscript easier to follow. To reiterate, i am in no good position to judge the genomic aspects of the paper, but concerning the functional data, i have no further issues. If length restrictions permit, i would still suggest to help th reader with the dense content by adding a few sentences to connect the different aspects of the paper and elaborate on the functional experiments in the last part (essentially Fig. 4).”

Response to the comments: Many thanks and we have revised accordingly in the revised version.

Reviewers' Comments:

Reviewer #1:

Remarks to the Author:

I have read the manuscript in its second version, which has been significantly improved. The authors have carefully addressed most of the comments, but I still have a major comment on the population structure analysis.

Major comment :

The authors presented in their answer to review the structure without related individuals. However, a substructure was still observed in the cultivated gene pool at $K > 4$, and at $K = 10$, a substructure was observed for the WRO population. Can the authors please provide:

- Plots of cross-validation as a function of the K values for the population structure analyses 1) with and 2). without related individuals?

-a PCA as Fig. 2 d for population structure analyses without related individuals highlighting the additional cluster observed for the cultivated rubber tree at $K = 6$

-a map as Fig. 2 a for population structure analyses without related individuals highlighting the additional cluster observed for the cultivated rubber tree at $K = 6$

-a PCA as Fig. 2 d for population structure analyses without related individuals highlighting the additional cluster observed for the WRO population at $K = 10$

-a map as Fig. 2 a for population structure analyses without related individuals highlighting the additional cluster observed for the WRO population at $K = 10$

-FST estimates among the clusters detected at $K = 6$ and at $K = 10$

These plots will help to determine whether $K = 4$ is the most relevant population structure. Finding the right population structure is crucial for downstream analyses: if one has a substructure in the cultivated rubber tree that has not been taken into account, then false positive signals can be found.

Reviewer #2:

Remarks to the Author:

I maintain that the authors failed to address my last concern in their revised version of the article. Indeed, I previously asked the authors to add complete statistics (fasta statistics, complete BUSCO score: complete, fragmented, duplicated, missing, Mercury completeness and QV, statistics of the gene catalogs) of other available assemblies in Table 1 (at least CATAS8-79 and GT1), but in this version these statistics are still spread over several additional tables.

As a reminder, it was the content of my previous review : "The article describes a genomic resource and potential users must have information on the quality of this resource compared to existing ones. Authors should provide a more comprehensive Table 1 with metrics (at least general statistics, BUSCO scores, Mercury completeness and QV) calculated on available genome assemblies."

Responses to the Reviewers' comments on the MS (NCOMMS-22-20274B)

We here gratefully thank all the reviewers for their critical comments. The manuscript is further improved. All the revisions are marked in red color. The followings are the point-to-point responses to the reviewers' comments on the MS (NCOMMS-22-20274B).

Reviewer1's major comment: "I have read the manuscript in its second version, which has been significantly improved. The authors have carefully addressed most of the comments, but I still have a major comment on the population structure analysis. Major comment :

The authors presented in their answer to review the structure without related individuals. However, a substructure was still observed in the cultivated gene pool at $K > 4$, and at $K = 10$, a substructure was observed for the WRO population. Can the authors please provide:

- (1) - Plots of cross-validation as a function of the K values for the population structure analyses 1) with and 2). without related individuals?
- (2) -a PCA as Fig. 2 d for population structure analyses without related individuals highlighting the additional cluster observed for the cultivated rubber tree at $K = 6$
- (3) -a map as Fig. 2 a for population structure analyses without related individuals highlighting the additional cluster observed for the cultivated rubber tree at $K = 6$
- (4) -a PCA as Fig. 2 d for population structure analyses without related individuals highlighting the additional cluster observed for the WRO population at $K = 10$
- (5) -a map as Fig. 2 a for population structure analyses without related individuals highlighting the additional cluster observed for the WRO population at $K = 10$
- (6) -FST estimates among the clusters detected at $K = 6$ and at $K = 10$

These plots will help to determine whether $K = 4$ is the most relevant population structure. Finding the right population structure is crucial for downstream analyses: if one has a substructure in the cultivated rubber tree that has not been taken into account, then false positive signals can be found."

Response to “provide (1)-Plots of cross-validation as a function of the K values for the population structure analyses 1) with and 2). without related individuals”:

We are very grateful for your very professional and meticulous guidance. Based on your kind suggestion, we plotted a cross-validation curve (a Built-in program of ADMIXTURE software) of the population with or without related individuals (please kindly refer to the figure 1a and b). The curve is relatively stable when $K > 4$. We also re-checked the likelihood value generated by fastStructure software (figure 1c). The result showed that the log-likelihood ratio was also stable when $K > 4$, suggesting four groups were the most relevant population structure in this study.

figure1 K value determination. (a) Cross-validation curve generated by Admixture software using 312 samples excluding 24 highly related individuals; (b) Cross-validation curve generated by Admixture software using 336 samples including 24 highly related individuals; (c) Log-Likelihood curve generated by fastStructure software using 312 samples excluding 24 highly related individuals.

Meanwhile, the source information of the wild rubber tree germplasms is well documented because the wild germplasms are collected from the three distinct states of Brazil, Acre, Rondonia and Mato Grosso by IRRDB (International Rubber Research and Development Board) in 1981. The three wild populations and cultivated population were also well displayed by using limited RFLP markers (Besse P, Seguin M, Lebrun P, Chevallier MH, Nicolas D, Lanaud C. Genetic Diversity among Wild and Cultivated Populations of *Hevea brasiliensis* Assessed by Nuclear RFLP Analysis. *Theor Appl Genet.* 1994;88: 199-207.).

Response to “provide (2) -a PCA as Fig. 2 d for population structure analyses without related individuals highlighting the additional cluster observed for the cultivated rubber tree at K= 6”:

It was provided in figure 2b. The result showed that the additional cluster (marked with E) was more related to cluster D, both of which were the substructure of clade Cul.

Response to “provide (3) -a map as Fig. 2 a for population structure analyses without related individuals highlighting the additional cluster observed for the cultivated rubber tree at K= 6”:

It was provided in figure 2a. The additional cluster marked with E included 20 cultivars from China, 2 cultivars from Malaysia, 1 cultivar from Thailand and 1 cultivar from Indonesia.

Response to “provide (4) -a PCA as Fig. 2 d for population structure analyses without related individuals highlighting the additional cluster observed for the WRO population at K= 10”:

It was provided in figure 3b. The result showed that the additional cluster (marked with B) was more related to cluster C, both of which were the substructure of clade WRO.

Response to “provide (5) - a map as Fig. 2 a for population structure analyses without related individuals highlighting the additional cluster observed for the WRO population at K= 10”:

It was provided in figure 3a. The germplasm in the additional cluster marked with B from the state of Rondonia, Brazil.

Response to “provide (6) -FST estimates among the clusters detected at K=6 and at K=10”:

The FST estimates among the clusters detected at K=6 and at K=10 were respectively provided in figure 2c and figure 3c. The minimum FST value was observed between the substructure and its relative clade, suggesting the low diversity between them.

figure 2 Population analysis of 312 samples (excluding 24 highly related individuals) by fastStructure software. (a) Displayed population structure at K=6, and the individual source of group E were labeled in map. (b) PCA plot of the first three components (PC1, PC2 and PC3), and the group E were circled. (c) FST values were calculated between spatial groups.

figure 3 Population analysis of 312 samples (excluding 24 highly related individuals) by fastStructure software. (a) Displayed population structure at K=10, and the individual source of group B were labeled in map. (b) PCA plot of the first three components (PC1, PC2 and PC3), and the group B were circled. (c) FST values were calculated between spatial groups.

We also noticed the unusual population structure at $K > 6$ analyzed by fastStructure. The substructure in Clu clade which cannot be deconstructed along with the increase in K values may lead to the doubt that there are some related or duplicated individuals in the population. We carefully re-checked the individual information and re-analyzed the population structure by ADMIXTURE software (admixture --cv=5 \$bimfile \$Kmer | tee log\$Kmer.out.). The calculation results and logs were updated to the website: <https://doi.org/10.5281/zenodo.7672530>. Results showed the individual information was correct, and a clearer population structure was obtained compared to previous fastStructure analysis (please kindly refer to the figure 4-5). In these new results, the number of substructures were in accordance with the K number, and the substructure in fastStructure results was now well displayed and further deconstructed from K=6 to K=10 (figure 4-5). The individuals of each group obtained by the two methods (fastStructure and ADMIXTURE) is consistent at K=4 (figure 4-5).

figure 4 Population structure analysis of 312 samples (excluding 24 highly related individuals) by fastStructure and Admixture softwares.

figure 5 Population structure analysis of all 336 samples by fastStructure and Admixture softwares.

We further re-analyzed both the PCA and map for population structure analyses at $K=6$, and calculated F_{ST} value with or without related individuals. The substructure mainly composed of Tjir1 offspring in cultivar without related individuals (figure 6) while the substructure mainly composed of PR107 offspring or Tjir1 offspring in cultivar with related individuals (figure 7). The minimum F_{ST} value was observed between the substructure within same clade (figure 6c and 7c).

figure 6 Population analysis of 312 samples (excluding 24 highly related individuals) by Admixture software. (a) Displayed population structure at $K=6$, and the individual source of substructure F were labeled in map. (b) PCA plot of the first three components (PC1, PC2 and PC3), and the substructure F were circled. (c) F_{ST} values were calculated between spatial groups.

figure 7 Population analysis of 336 samples (including 24 highly related individuals) by Admixture software. (a) Displayed population structure at K=6, and the individual source of substructure E and F were labeled in map. (b) PCA plot of the first three components (PC1, PC2 and PC3), and the substructure E and F were circled. (c) FST values were calculated between spatial groups.

We then re-analyzed both the PCA and map for population structure analyses (Admixture software) at k=10, and calculated FST value with or without related individuals (Figure 8 and 9). The substructure mainly composed of PR107 offspring or Tjir1 offspring in cultivar with related individuals while the substructure in clade WAC, WRO, WMG were associated with the geographic location. The minimum FST value was observed between the substructures within the same clade (figure 8c and 9c).

figure 8 Population analysis of 312 samples (excluding 24 highly related individuals) by Admixture software. (a) Displayed population structure at K=10, and the individual source of substructure B, C, E, F were labeled in map. (b) PCA plot of the first three components (PC1, PC2 and PC3), and all substructures were circled. (c) FST values were calculated between spatial groups.

figure 9 Population analysis of 336 samples (including 24 highly related individuals) by Admixture software. (a) Displayed population structure at K=10, and the individual source of substructure B, C, E were labeled in map. (b) PCA plot of the first three components (PC1, PC2 and PC3), and all substructures were circled. (c) FST values were calculated between spatial groups.

Taken together, the above analyses supported that K=4 is the most relevant population structure, and the results by fastStructure and ADMIXTURE were consistent at K = 4. Considering that the results by ADMIXTURE software was clearer than fastStructure software especially the clearly deconstructed substructures within the same clade, we used the data obtained by the ADMIXTURE software in the revised version. In addition, considering that there were no admixtures, such as highly related individuals and duplicates, we use 336 population in the revised version.

Reviewer2's major comment: "I maintain that the authors failed to address my last concern in their revised version of the article. Indeed, I previously asked the authors to add complete statistics (fasta statistics, complete BUSCO score: complete, fragmented, duplicated, missing, Mercury completeness and QV, statistics of the gene catalogs) of other available assemblies in Table 1 (at least CATAS8-79 and GT1), but in this version these statistics are still spread over several additional tables.

As a reminder, it was the content of my previous review: "The article describes a genomic resource and potential users must have information on the quality of this resource compared to existing ones. Authors should provide a more comprehensive Table 1 with metrics (at least general statistics, BUSCO scores, Mercury completeness and QV) calculated on available genome assemblies."

Response: We are very sorry for misunderstanding your concern. In this revised version, we integrated more detailed information on the sequence statistics, the feature of genome assembly and gene annotation of CATAS8-79 in Table 1 in comparison with those of GT1 genome, both of which were at chromosome-level. Considering some information of GT1 was not available in the published paper (Gap number, Mercury completeness, Mercury quality value, Max gene length, Max transcript length, etc), we re-analyzed the GT1 assembly and supplied the information in Table 1. The available data from other versions of rubber tree genome assembly was listed in Supplementary Table 4.

Table 1 Statistics of rubber tree clone CATAS8-79 and GT1 genome assemblies.

	CATAS8-79	GT1 (Liu et al. 2020)
Sequence statistics		
Next-Generation Sequencing (Gb)	85.58	348.14
PacBio (Gb)	192.00	161.86
Hi-C (Gb)	110.00	119.63
BioNano (Gb)	206.85	n.a.
Feature of genome assembly		
Total length of assembly (Gb)	1.58	1.56
Total no. of contigs	1,058	16,023
Contig N50 (Mb)	11.21	0.15
Longest contig (Mb)	56.83	1.81
Total no. of scaffold	776	600
Scaffold N50 (Mb)	94.82	84.02
Longest scaffold (Mb)	112.33	1.81
Total no. of anchored contigs	300	15,324
Gap number	282	15,306
Total length of chromosome-level assembly (Gb)	1.55	1.44
Anchor rate	98.17%	92.40%
Mercury completeness	82.11%	82.37%
Mercury quality value	28.92	24.62
Complete BUSCOs	1,546 (95.80%)	1,509 (93.50%)
Complete and single-copy BUSCOs	1,377 (85.30%)	1,349 (83.60%)
Complete and duplicated BUSCOs	169 (10.50%)	160 (9.90%)
Fragmented BUSCOs	34 (2.10%)	44 (2.70%)
Missing BUSCOs	34 (2.10%)	61 (3.80%)
Total BUSCO groups searched	1,614 (100.00%)	1,614 (100.00%)
Gene annotation		
No. of gene models	38,595	44,187
Percentage of gene length in genome	13.15%	11.10%
Max gene length (bp)	75,969	72,122
Average gene length (bp)	5,282.54	3,918.40
No. of transcripts	82,641	102,235
Max transcript length (bp)	21,964	10,920
Average transcript length (bp)	1,680.61	1,065.00
Average CDS length (bp)	1,275.52	1,140.10
Average exon length (bp)	217.09	222.08
Average intron length (bp)	552.46	672.1
Average exon number per gene	7.74	5.13
Repeat sequence length (Mb)	1,180.12	1,042.40
Repeats percentage of assembly	75.92%	70.81%

Reviewers' Comments:

Reviewer #1:

Remarks to the Author:

The authors replied to my comments. However, I am sorry, but I still believe that the authors must remove the closely related individuals for fastSTRUCTURE analyses (i.e., offspring of Tjr1 and Pr107, taking only one representative of each clone). Highly genetically related individuals can lead to the detection of spurious clusters (as the authors show at $K=6$ when related individuals are kept, Figure 2 answers to review). Therefore, the argument that K stays stable for $K>4$ without and with closely related individuals is invalid in that context. The authors should remove the closely related individuals.

Once the closely related individuals are removed, no additional clusters can be detected for $K>10$ (Figure 4 answer to review). For $K=10$, another cluster is observed in the WRO population, and no substructure is observed in the CUL population. Figure 1 of the answer to review is interesting in this context: when you look at Figure 1c, there is a peak at $K=4$ and another at $K=10$.

Thus, the analyses shown in Figure 3 should be the ones presented in the main manuscript, and the solution at $K=10$ (therefore assuming that the WRO population splits in two). In addition, the follow-up selection analyses for the WRO populations should be run separately for WRO1 and WRO2 to avoid any bias in the site frequency spectrum estimates.

Reviewer #2:

Remarks to the Author:

I have read the revised manuscript and my main comment has been addressed. I noticed a typo in Table 1, I think the size of the largest scaffold for the GT1 assembly is wrong.

Responses to the Reviewers' comments on the MS (NCOMMS-22-20274C)

Dear Reviewers,

We here gratefully thank your critical comments. The manuscript is further improved. All the revised are marked in red color. The followings are the point-to-point responses to the reviewers' comments on the MS (NCOMMS-22-20274C).

The Reviewer1's major comments: "The authors replied to my comments. However, I am sorry, but I still believe that the authors must remove the closely related individuals for fastSTRUCTURE analyses (i.e., offspring of Tjr1 and Pr107, taking only one representative of each clone). Highly genetically related individuals can lead to the detection of spurious clusters (as the authors show at $K=6$ when related individuals are kept, Figure 2 answers to review). Therefore, the argument that K stays stable for $K>4$ without and with closely related individuals is invalid in that context. The authors should remove the closely related individuals. Once the closely related individuals are removed, no additional clusters can be detected for $K>10$ (Figure 4 answer to review). For $K=10$, another cluster is observed in the WRO population, and no substructure is observed in the CUL population. Figure 1 of the answer to review is interesting in this context: when you look at Figure 1c, there is a peak at $K=4$ and another at $K=10$. Thus, the analyses shown in Figure 3 should be the ones presented in the main manuscript, and the solution at $K=10$ (therefore assuming that the WRO population splits in two). In addition, the follow-up selection analyses for the WRO populations should be run separately for WRO1 and WRO2 to avoid any bias in the site frequency spectrum estimates. "

Response to the major comments:

Thanks for your kindly suggestion.

(1) We have revised the Fig. 2 with solution at K=10 in the main manuscript according to your suggestion.

We removed the cultivars with rubber tree primary clone Tjir1's or PR107's pedigree in clade Cul and re-constructed the population structure of 245 individuals by using fastStructure software (Figure 1a). An inflection point was observed in log-Likelihood curve at K=4 (See Figure 1a and b below). Indeed, "Choice of an appropriate value for K is a notoriously difficult statistical problem. It seems to us that this choice should be guided by knowledge of a population's history" (Alexander D H, Novembre J, Lange K. Fast model-based estimation of ancestry in unrelated individuals. *Genome Research*, 2009, 19(9):1655). The population's history of rubber tree in the present study was very clear: the cultivars were derived from the germplasms collected in Boim of Brazil by Henry Wickham in 1876, and the wild germplasms were respectively collected from three states (Acre, Rondonia, and Mato Grosso) of Brazil by IRRDB in 1981. Taken together, it was reasonable to separate the population into main four clades.

We re-performed population genetic analysis including PCA, FST, PAI, and LD. There was little difference in PCA, FST, PAI, and LD between the population with and without the individuals with rubber tree primary clone Tjir1's or PR107's pedigree (See Figure 2 below).

In summary, we have revised the Fig. 2 in the main manuscript according to your suggestion.

(2) Selection analysis was performed by using *Fst* between WRO1 and WRO2. Data were deposited to the website: <https://doi.org/10.5281/zenodo.7672530>. It is noted that the aim of selection analysis (such as FST, PAI, XP-CLR) is to uncover the molecular signals associated with quantitative traits which are selected during domestication. Therefore, the selected regions between WRO1 and WRO2 should be caused by diverse location of these wild germplasms rather than artificial selection by breeders.

Figure 1 Population structure analysis of 245 individuals after removing the cultivars with rubber tree primary clone Tjir1' or PR107' pedigree (a) and K value determination by Log-Likelihood curve (b) generated by fastStructure software.

Figure 2 Comparison of population genetic analysis between 245 individuals with exclusion of 35 offspring of Tjir1 and PR107 from 280 individuals and all the 280 out of the 335 individuals by removing the 55 admixed individuals based on phylogenetic analysis.

The Reviewer2's comment: "I have read the revised manuscript and my main comment has been addressed. I noticed a typo in Table 1, I think the size of the largest scaffold for the GT1 assembly is wrong."

Response to review:

Thanks. We have corrected it in the revised version. The largest scaffold for the GT1 assembly is 103.98 Mb rather than 1.81 Mb.

Reviewers' Comments:

Reviewer #1:

Remarks to the Author:

The authors have replied to my comments.

Point-by-point response to reviewers' comments on MS-22-20274D

Reviewer #1' comment: "The authors have replied to my comments."

Response: Thanks.